# LARGE BRAIN MODEL FOR LEARNING GENERIC REPRESENTATIONS WITH TREMENDOUS EEG DATA IN BCI

**Wei-Bang Jiang[1], Li-Ming Zhao[2]* & Bao-Liang Lu[12]***

[1]Shanghai Jiao Tong University   [2]Shanghai Emotionhelper Technology Co., Ltd.

`935963004@sjtu.edu.cn,liming.zhao@emotionhelper.com,bllu@sjtu.edu.cn`

## ABSTRACT

The current electroencephalogram (EEG) based deep learning models are typically designed for specific datasets and applications in brain-computer interaction (BCI), limiting the scale of the models and thus diminishing their perceptual capabilities and generalizability. Recently, Large Language Models (LLMs) have achieved unprecedented success in text processing, prompting us to explore the capabilities of Large EEG Models (LEMs). We hope that LEMs can break through the limitations of different task types of EEG datasets, and obtain universal perceptual capabilities of EEG signals through unsupervised pre-training. Then the models can be fine-tuned for different downstream tasks. However, compared to text data, the volume of EEG datasets is generally small and the format varies widely. For example, there can be mismatched numbers of electrodes, unequal length data samples, varied task designs, and low signal-to-noise ratio. To overcome these challenges, we propose a unified foundation model for EEG called Large Brain Model (LaBraM). LaBraM enables cross-dataset learning by segmenting the EEG signals into EEG channel patches. Vector-quantized neural spectrum prediction is used to train a semantically rich neural tokenizer that encodes continuous raw EEG channel patches into compact neural codes. We then pre-train neural Transformers by predicting the original neural codes for the masked EEG channel patches. The LaBraMs were pre-trained on about 2,500 hours of various types of EEG signals from around 20 datasets and validated on multiple different types of downstream tasks. Experiments on abnormal detection, event type classification, emotion recognition, and gait prediction show that our LaBraM outperforms all compared SOTA methods in their respective fields. Our code is available at `https://github.com/935963004/LaBraM`.

## 1 INTRODUCTION

Electroencephalography (EEG) is a method to record an electrogram of the spontaneous electrical activity of the brain. It is typically non-invasive, with the EEG electrodes placed along the scalp using the international 10–20 system. EEG signals can be formulated as a matrix of real numbers $X \in \mathbb{R}^{C \times T}$, where $C$ is the number of EEG electrodes (channels) that may vary depending on the acquisition equipment used, and $T$ represents the total number of samples, which is related to the collection time and sampling rate. As highly objective physiological signals, EEG has demonstrated remarkable potential in seizure epilepsy classification (Boonyakitanont et al., 2020), acute stress detection (Sharma et al., 2022), sleep stage classification (Aboalayon et al., 2016), motor imagery recognition (Amin et al., 2019), abnormal identification (Roy et al., 2019), emotion analysis (Suhaimi et al., 2020), and auditory attention detection (Biesmans et al., 2016).

Numerous deep learning models have been proposed to address the aforementioned tasks in their respective fields. Some works apply convolutional neural networks (CNN) across and within raw EEG channels to encode spatial and temporal features (Lawhern et al., 2018), while others preprocess the data using short-time Fourier transform (STFT) and employ Graph Neural Network (GNN)

---

*Li-Ming Zhao and Bao-Liang Lu are co-corresponding authors.

on the resulting spectrograms to obtain semantic features of brain area links (Song et al., 2018). Researchers also segment the signal and use a CNN segment encoder with a downstream sequence model such as recurrent neural networks (RNN) to capture temporal dynamics (Xu et al., 2020). These models primarily focus on EEG samples that adhere to specific task formats, mainly because the equipment used to collect EEG differs between datasets, which introduces mismatched channels and variable lengths. Meanwhile, EEG data collection is quite expensive, which makes it challenging to build large EEG datasets specifically designed for a particular task. To prevent overfitting, the parameters of these models need to be regulated, which in turn hampers the model's ability to learn EEG expressions and limits its generalizability. Consequently, we discovered that current EEG models are typically proprietary and lack the capacity to perform cross-task learning.

Recently, we have been impressed by the capabilities of LLMs (Ouyang et al., 2022; Wei et al., 2022). Specifically, Transformer-based models have demonstrated promising results in natural language processing tasks, which highlights the potential of self-supervised pre-training as a means for harnessing large-scale data. These masked language modeling tasks involve randomly masking some proportion of tokens within a text and then recovering the masked tokens based on the Transformer encoding results of the corrupted text. Motivated by these methods, we propose to apply reconstruction ideas to pre-train neural Transformers. However, it is a daunting task to directly apply LLM-style pre-training to EEG data. The challenges are summarized as follows:

**1) Lack of sufficient EEG data.** The acquisition of EEG data is significantly challenging compared to natural language and image data. Moreover, the annotation of EEG data usually requires a lot of effort on the part of experts in the corresponding field, thus leading to the fact that only small labeled datasets exist for specific tasks in BCI, where EEG signals are often collected from a small number of participants, typically less than tens of hours in duration. As a result, there is currently no single EEG dataset that is large enough to support the training of LEMs. It remains problems **Q1:** *how to utilize large-scale unlabeled EEG data?* and **Q2:** *how much data is needed to train LEMs?*.

**2) Diverse configurations of EEG collection.** Despite the availability of the international 10-20 system to ensure standardization in EEG testing, users may choose to collect data using EEG caps with different electrode numbers or patch electrodes based on their practical application needs. Thus, how to handle the diverse formats of EEG data in order to match the input units of neural Transformers remains a significant research endeavor.

**3) Lack of effective EEG representation learning paradigm.** Low signal-to-noise ratio (SNR) and different types of noise are the greatest challenges. Additionally, balancing temporal and spatial characteristics is crucial for effective EEG representation learning. Despite the availability of various deep learning-based EEG representation learning paradigms, such as CNN, RNN, and GNN, for raw EEG data, many researchers still prefer to design artificial EEG features due to these challenges.

In this paper, our objective is to devise a versatile large EEG model that can efficiently handle diverse EEG datasets with varying channels and lengths. By utilizing unsupervised training on a substantial amount of EEG data, we envision the model to possess universal EEG data comprehension capabilities, enabling it to quickly adapt to various EEG downstream tasks. We collected over 2,500 hours of diverse EEG data across various tasks and formats from about 20 datasets. These datasets were primarily obtained from publicly available EEG datasets, as well as our own collected EEG data. Raw EEG signals were first segmented into EEG channel patches to deal with the issues of variant electrodes and time length. Vector-quantized neural spectrum prediction is used to train a semantically rich neural tokenizer to generate neural vocabulary. Specifically, the tokenizer was trained by predicting the Fourier spectrum of the original signal. During pre-training, part of EEG patches are masked while the objective of the neural Transformer is to predict masked tokens from visible patches. We pre-trained three models with varying parameter sizes, ranging from 5.8M to 369M, which are the largest models in BCI ever, and fine-tuned them on four distinct types of downstream tasks encompassing both classification and regression. The contributions of this work are summarized as follows:

- **Large-scale EEG pre-training.** We collected and pre-trained a large-scale neural Transformer model on more than 2,500 hours of diverse EEG data. As far as we know, this is the first time such extensive and varied datasets have been utilized for EEG pre-training.

- **Being compatible with various EEG configurations.** LaBraMs are unified models that are able to handle EEG signals with various channels and time lengths with the assistance of the flexible

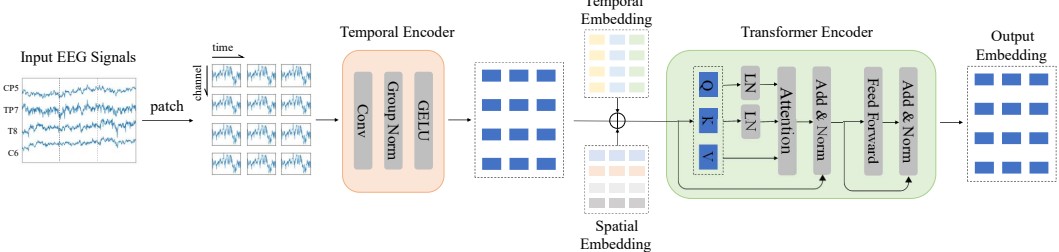

Figure 1: The overall architecture of LaBraM, i.e., neural Transformer. All input EEG signals will first be segmented into EEG patches through a fixed-length time window, and then a temporal encoder will be applied to each patch to extract temporal features. Afterward, temporal and spatial embeddings are added to the patch features to carry temporal and spatial information. At last, the sequence of embeddings is passed into the Transformer encoder by patch-wise attention to obtain the final output.

spatial and temporal embeddings. Hence, one pre-trained LaBraM can adapt to any downstream dataset with different configurations.

- **Effective EEG representation learning.** The utilization of the neural Transformer allows the model to effectively capture both temporal and spatial features of EEG signals with varying channels and lengths, making it suitable for a wide range of downstream tasks in EEG analysis. We further define a neural codebook that offers a compact, versatile, and meaningful representation of EEG signals. We resolve **Q1** by leveraging this codebook to pre-train LaBraM by masked EEG modeling. The empirical performance demonstrates the effectiveness of our proposed method and paves the way for further development in aligning this codebook with natural language.

- **Comprehensive experiments on downstream datasets.** We evaluate our LaBraMs on four representative downstream tasks in BCI, where they surpass all SOTA methods by a large margin. Additionally, we conduct experiments to answer **Q2** by scaling the pre-training data size and conclude the amount of pre-training data required for models of different sizes in Section 3.6.

## 2 METHOD

In this section, we detail the whole framework of LaBraM. We first formulate the multi-channel EEG signals as $X \in \mathbb{R}^{C \times T}$, where $C$ is the number of EEG electrodes (channels) and $T$ is the total timestamps. The electrode set of $X$ is formulated as $\mathcal{C}_X = \{c_{i_1}, c_{i_2}, ..., c_{i_C}\}$, where $\mathcal{C}_X \subseteq \mathcal{C} = \{c_1, c_2, ..., c_{|\mathcal{C}|}\}$ and $\mathcal{C}$ is the universal set of channels in the international 10-20 system.

### 2.1 MODEL ARCHITECTURE

We introduce the neural Transformer, a general architecture for decoding EEG signals that can deal with any input EEG signals with arbitrary number of channels and time length, as illustrated in Figure 1. The key operation for achieving this is segmenting the EEG signals into patches, inspired by patch embeddings in images (Dosovitskiy et al., 2021). Assume that the timestamp for each sample is $t$ and the stride is $s$. $X$ can be segmented into $\lfloor \frac{T-t}{s} \rfloor + 1$ samples, and each sample $\boldsymbol{x} \in \mathbb{R}^{C \times t}$. We use a $w$-length window without overlap to segment each EEG channel into patches, obtaining $\boldsymbol{x} = \{x_{c_{i_j},k} \in \mathbb{R}^w | j = 1, 2, ..., C, k = 1, 2, ..., \lfloor \frac{t}{w} \rfloor\}$. The total number of the patches $\boldsymbol{x}$ is $|\boldsymbol{x}| = C \lfloor \frac{t}{w} \rfloor$.

**Temporal Encoder.** As EEG is of high resolution in the temporal domain, it is vital to extract temporal features before patch-wise interaction by self-attention. We employ a temporal encoder which consists of several temporal convolution blocks to encode each EEG patch into a patch embedding. The temporal convolution block is composed of a 1-D convolution layer, a group normalization layer (Wu & He, 2018), and a GELU activation function (Hendrycks & Gimpel, 2016). We denote the output patch embeddings from the temporal encoder as

$$\boldsymbol{e} = \{e_{c_{i_j},k} \in \mathbb{R}^d | j = 1, 2, ..., C, k = 1, 2, ..., \lfloor \frac{t}{w} \rfloor\}, \tag{1}$$

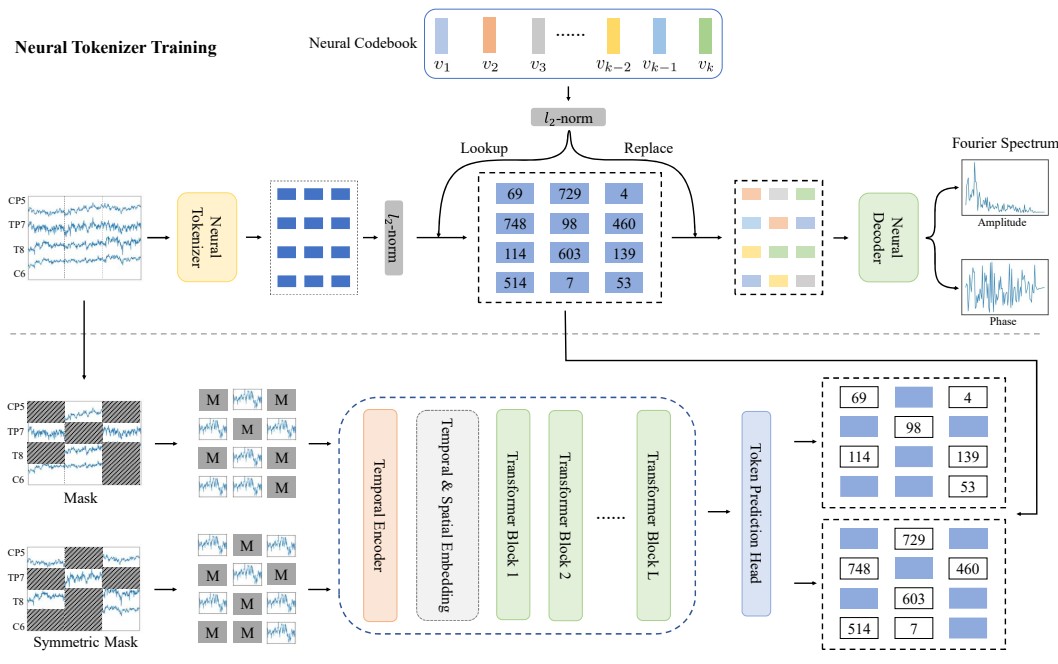

Figure 2: Overview of neural tokenizer training and LaBraM pre-training. **Up**: We train a neural tokenizer to discretize EEG signals into discrete neural tokens by reconstructing the Fourier spectrum. **Down**: During pre-training, part of EEG patches are masked while the objective is to predict masked tokens from visible patches.

where $d$ is the dimension of the embeddings.

**Temporal & Spatial Embedding.** In order to enable the model to be aware of the temporal and spatial information of patch embeddings, we initialize a temporal embedding list $TE = \{te_1, te_2, ..., te_{tmax}\}$ and a spatial embedding list $SE = \{se_1, se_2, ..., se_{|\mathcal{C}|}\}$, both of which are $d$-dimension and are set learnable during training. Note that $tmax$ is the hyperparameter determining the maximum number of time patches and $\lfloor \frac{t}{w} \rfloor \leq tmax$. Meanwhile, for each channel $c_i$, we can find its corresponding spatial embedding $se_i$ in the spatial embedding list $SE$. Thus, given one arbitrary output embedding $e_{c_{i_j},k}$ in Equation 1 from the temporal encoder, we add the corresponding temporal and spatial embeddings to it:

$$\{e_{c_{i_j},k} + te_k + se_{i_j} | j = 1, 2, ..., C, k = 1, 2, ..., \lfloor \frac{t}{w} \rfloor\}, \tag{2}$$

where temporal and spatial embeddings act as absolute position encoding.

**Transformer Encoder.** Finally, the sequence of embeddings will be directly fed into the Transformer encoder (Vaswani et al., 2017). To make the training of Transformer more stable and efficient, we incorporate some modifications (Dehghani et al., 2023). First, we add layer normalization to the queries and keys before the dot-product attention mechanism, which avoids over-large values in attention logits:

$$\text{Attention}(Q, K, V) = softmax(\frac{\text{LN}(Q)\text{LN}(K)^T}{\sqrt{d_{head}}})V, \tag{3}$$

where $d_{head}$ is the dimension of one head in the multi-head attention and LN denotes the layerNorm (Ba et al., 2016). Next, we omit the bias term in QKV computations, which accelerates the training without performance degradation. For downstream tasks, we use average pooling on the output embeddings followed by task-specific prediction heads.

## 2.2 Neural Tokenizer Training

Prior to pre-training LaBraM through masking and prediction, we need to tokenize the EEG into discrete tokens. We propose the vector-quantized neural spectrum prediction, which is trained by predicting the Fourier spectrum, as shown in Figure 2. The key components are the neural tokenizer which encodes EEG samples into patch representations and the neural decoder which decodes the Fourier spectrum from neural embeddings. The idea is basically inspired by VQ-VAE (Van Den Oord et al., 2017) which encodes images into discrete latent representations.

**Neural Tokenizer.** We define a neural codebook $\mathcal{V} = \{v_i | i = 1, ..., K\} \in \mathbb{R}^{K \times D}$, where $K$ is the number of the discrete neural embeddings and $D$ is the dimensionality of each embedding. Given an EEG signal sample $x$, the neural tokenizer whose backbone is just described in Section 2.1 first encode it to patch representations $\boldsymbol{p} = \{p_i | i = 1, ..., N\}$, where $N = C \lfloor \frac{t}{w} \rfloor$. After that, we utilize a quantizer to quantize all the patch representations into the neural codebook embeddings. The codebook looks up the nearest neighbor of each patch $p_i$ in the neural codebook $\mathcal{V}$. This procedure can be formulated as

$$z_i = \arg \min_j \|\ell_2(p_i) - \ell_2(v_i)\|_2, \tag{4}$$

where $\ell_2$ represents $\ell_2$ normalization and $z_i$ is the quantized vector after the quantizer. This is equivalent to finding the closest neural embedding by cosine similarity and such $\ell_2$ normalization improves the codebook utilization (Peng et al., 2022).

**Fourier Spectrum Prediction.** Unlike images that are of high signal-to-noise ratio, EEG signals are of low signal-to-noise ratio and have characteristics of apparent stochasticity, nonstationarity, and nonlinearity nature, which make it hard to reconstruct the original signals well (Moss et al., 2004). In our previous experiments, the loss fails to converge while directly reconstructing raw EEG signals. Instead, the frequency and phase distribution from the Fourier spectrum of EEG signals reveals the underlying neurophysiological activities of the brain (Wu et al., 2022). Therefore, we propose to reconstruct the amplitude and phase from discrete neural tokens for training the neural tokenizer and neural decoder. For an EEG patch $x_{c,k} = [x[1], x[2], ..., x[w]]$ of channel $c$ and time $k$ in a sample $x$, we apply the Discrete Fourier Transform (DFT) as follows

$$\tilde{x}_{c,k}^m = \sum_{n=1}^{N} x[n] \exp(-\frac{2\pi j}{N} mn), \tag{5}$$

where $m \in [1, N]$ and $j$ is the imaginary unit. We rewrite Equation 5 using Euler's formula as

$$\tilde{x}_{c,k}^m = \sum_{n=1}^{N} x[n] \cos(\frac{2\pi}{N} mn) - j x[n] \sin(\frac{2\pi}{N} mn). \tag{6}$$

Note that $\tilde{x}_{c,k}^m$ indicates the spectrum of the sequence at frequency $\omega_m = \frac{2\pi m}{N}$. Consequently, the amplitude and phase can be calculated as

$$A^m = \sqrt{Re(\tilde{x}_{c,k}^m)^2 + Im(\tilde{x}_{c,k}^m)^2}, \tag{7}$$

$$\phi^m = \arctan(\frac{Im(\tilde{x}_{c,k}^m)}{Re(\tilde{x}_{c,k}^m)}), \tag{8}$$

where $Re$ and $Im$ stand for the real and imaginary parts of a complex number. It is worthwhile to mention that we adopt z-score normalization to normalize $A^m$ and $\phi^m$ within a sample for stable convergence.

After being tokenized by the quantizer, the normalized discrete neural embeddings $\{\ell_2(v_{z_i}) | i = 1, ..., N\}$ are passed into the neural decoder that comprises several Transformer blocks. The output representations are aggregated by average pooling followed by two specific prediction heads to regress the spectrum amplitude $o^A$ and phase $o^\phi$, respectively. The mean squared error (MSE) loss is utilized to guide the prediction. Ultimately, the total loss for training the vector-quantized neural spectrum prediction is defined as

$$\mathcal{L}_T = \sum_{\boldsymbol{x} \in \mathcal{D}} \sum_{i=1}^{N} \|o_i^A - A_i\|_2^2 + \|o_i^\phi - \phi_i\|_2^2 + \|\boldsymbol{sg}(\ell_2(p_i)) - \ell_2(v_{z_i})\|_2^2 + \|\ell_2(p_i) - \boldsymbol{sg}(\ell_2(v_{z_i}))\|_2^2, \tag{9}$$

where $\mathcal{D}$ is all EEG data and $\boldsymbol{sg}$ represents the stop-gradient operation that is defined as an identity at the forward pass and has zero gradients. To make the codebook update more stable, we employ the exponential moving average strategy (Van Den Oord et al., 2017).

## 2.3 Pre-training LaBraM

**Masked EEG Modeling.** To enforce LaBraM learning generic representations with tremendous EEG data, we propose masked EEG modeling. The whole procedure is presented in Figure 2. As formulated in Section 2.1, given an EEG sample $\boldsymbol{x}$, the temporal encoder first transforms it to patch embeddings $\boldsymbol{e} = \{e_i | i = 1, ..., N\}$. We randomly generate a mask $\mathcal{M} = \{m_i | i = 1, ..., N\}$ where $m_i \in \{0, 1\}$ with $r$ proportion of $m$ is 1. After that, we replace the masked patches of $x$ with the learnable mask token $\boldsymbol{e}_M \in \mathbb{R}^d$. The corrupted EEG patches can be denoted as $\boldsymbol{e}^{\mathcal{M}} = \{e_i : m_i = 0 | i = 1, ..., N\} \cup \{e_M : m_i = 1 | i = 1, .., N\}$, which will be added by temporal and spatial embeddings, and then fed into Transformer encoder. We denote the output hidden vectors as $\boldsymbol{h} = \{h_i | i = 1, ..., N\}$, which are used to predict the corresponding neural tokens through a linear classifier:

$$p(\boldsymbol{v}'|\boldsymbol{e}^{\mathcal{M}}) = softmax(\text{Linear}(\boldsymbol{h})). \tag{10}$$

Our objective training loss is

$$\mathcal{L}_{\mathcal{M}} = -\sum_{\boldsymbol{x} \in \mathcal{D}} \sum_{m_i = 1} \log p(v_i | \boldsymbol{e}^{\mathcal{M}}). \tag{11}$$

**Symmetric Masking.** We further propose a symmetric masking strategy to improve training efficiency. We calculate the inverse of the generated mask $\mathcal{M}$, obtaining $\tilde{\mathcal{M}} = \{\sim m_i | i = 1, ..., N\}$. Similarly, we use the new mask $\tilde{\mathcal{M}}$ to perform the masked EEG modeling, obtaining the masked EEG prediction loss $\mathcal{L}_{\mathcal{M}}^{sym}$. The motivation is from two aspects: 1) Since we introduce the neural tokenizer, there will be an extra computation overhead, i.e., one forward pass for each EEG sample. Thus, the symmetric masking reuses the same discrete representations, thus improving training efficiency. 2) The symmetric masking provides more masking perspectives in one batch, increasing the data divergency. This simple strategy boosts downstream performance as demonstrated in Appendix I.

Finally, the overall training objective for pre-training LaBraM is

$$\mathcal{L} = \mathcal{L}_{\mathcal{M}} + \mathcal{L}_{\mathcal{M}}^{sym}. \tag{12}$$

## 3 Experiments

### 3.1 Evaluation Datasets

We systematically evaluate our LaBraM on the following downstream datasets:

- **TUAB** (abnormal detection) (Obeid & Picone, 2016): A corpus of EEGs which are 23-channel and sampled at 256 Hz. All data have been annotated as normal or abnormal. There are total 409,455 10-second samples that we use for binary classification to predict normal/abnormal.
- **TUEV** (event type classification) (Obeid & Picone, 2016): This corpus is a subset of TUEG that contains annotations of EEG segments as one of six classes: (1) spike and sharp wave (SPSW), (2) generalized periodic epileptiform discharges (GPED), (3) periodic lateralized epileptiform discharges (PLED), (4) eye movement (EYEM), (5) artifact (ARTF) and (6) background (BCKG). The EEG signals contain 23 channels at 256 Hz and are segmented into 112,491 5-second samples.

More experimental results on other BCI tasks can be found in Appendix F.

### 3.2 Experiment Setup

**Model Variants**. We devise three different configurations of LaBraM: LaBraM-Base, LaBraM-Large, and LaBraM-Huge. The number of parameters is 5.8M for LaBraM-Base, 46M for LaBraM-Large, and 369M for LaBraM-Huge, respectively, which is increased by enlarging the depth of

the Transformer encoder and hidden sizes. More details of the architecture settings are listed in Appendix C. Unless otherwise specified, the results are from LaBraM-Base in this paper.

The time window $w$ of a patch is set to 200 (1 second). To ensure stable computing resource usage, the number of patches (sequence length) is limited to 256. That means, for example, the time length of EEG with 64 (32) channels is set to 4 (8) seconds. As for the window stride (data stride), it is set to 4 seconds in order to cover all training data as well as boost the training speed.

**Pre-training & Fine-tuning**. For pre-training LaBraM and the vector-quantized neural spectrum prediction, we collect a total time of over 2,500 hours from public datasets and our self-collected data as described in Appendix D. Note that the four downstream datasets are excluded from the pre-training datasets. For the data splitting of TUAB and TUEV, we strictly follow the same strategy as BIOT (Yang et al., 2023a) to compare all methods fairly. Specifically, as the training and test separation is provided by the datasets, we divide the training patients into training and validation groups by 80% and 20%, respectively. We employ binary cross-entropy (BCE) loss for TUAB (binary classification) and cross-entropy loss for TUEV (multi-class classification), respectively. Our experiments are conducted on eight A800 GPUs by Python 3.11.4 and PyTorch 2.0.1 + CUDA 11.8. The best models are trained based on the training set, selected from the validation set, and finally evaluated on the test set. We report the average and standard deviation values on five different random seeds to obtain comparable results. (see Appendix C for more detailed hyperparameters)

**Preprocessing**. We only employ very little of the necessary preprocessing. We first filter the EEG signals between 0.1 Hz and 75 Hz to remove low-frequency noise. Then, a notch filter of 50 Hz is applied to avoid power-line interference. Finally, all EEG signals are resampled to 200 Hz. As the range of EEG value is typically between -0.1 mV to 0.1 mV, we normalize it by setting the unit to 0.1 mV to guarantee the value mainly between -1 to 1.

**Baselines & Metrics**. The baselines are from Yang et al. (2023a), where we choose the best results to compare with. We use the following metrics for comparison: 1) **Balanced Accuracy**: the average of recall on each class, which is utilized for both binary and multi-class classification. 2) **AUC-PR**: area under the precision-recall curve for binary classification. 3) **AUROC**: area under the receiver operating characteristic curve, which is used for binary classification as well. 4) **Cohen's Kappa**: a measure of agreement between categorical variables $X$ and $Y$, which is calculated from the observed and expected frequencies on the diagonal of a square contingency table. It is used for multi-class classification. 5) **Weighted F1**: A harmonic mean of the precision and recall, where the relative contribution of precision and recall to the F1 score are equal. We use it to evaluate multi-class classification. We set AUROC as the monitor score for binary classification and Cohen's Kappa as the monitor score for multi-class classification.

### 3.3 PRE-TRAINING VISUALIZATION

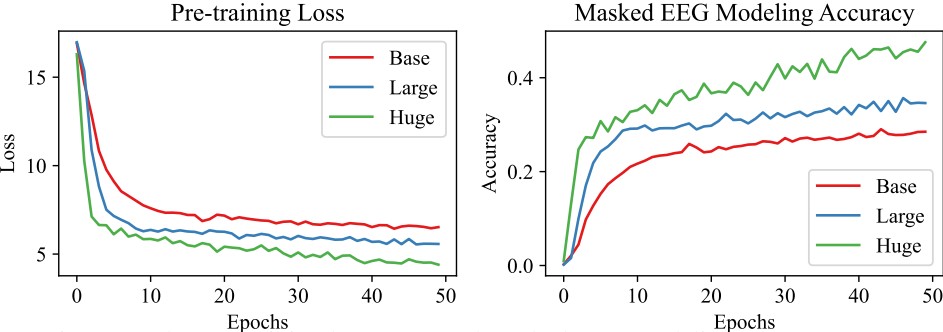

Figure 3: The pre-training loss curve and masked EEG modeling accuracy curve.

Figure 3 compares the convergence curves of the total pre-training loss and masked EEG modeling accuracy between the base, large, and huge models. We observe that a larger model with more parameters can converge to a smaller loss and higher accuracy. Notably, the loss of the huge model seems to have an obvious downward trend while the accuracy tends to increase if we train it longer. This observation suggests scaling up the model size has the potential to obtain better performance.

## 3.4 COMPARISON WITH STATE-OF-THE-ART

Table 1 and Table 2 present the results of state-of-the-art baselines as well as LaBraM from TUAB and TUEV. The results demonstrate that our LaBraM-Base model outperformed all baselines on various evaluation metrics for both tasks. Particularly in the more challenging multi-class classification task of TUEV, our model achieved a significant improvement in performance. In our own model, we observed that as the number of model parameters increased, the LaBraM-Huge model performed the best, followed by the LaBraM-Large model and then the LaBraM-Base model. We attribute this good performance to the increase in pre-training data volume and model parameters. We believe that with sufficient data volume, large-scale EEG models can learn more generalizable EEG patterns, leading to improved performance on a wide range of downstream tasks in EEG analysis.

Table 1: The results of different methods on TUAB.

| Methods | Model Size | Balanced Accuracy | AUC-PR | AUROC |
|---|---|---|---|---|
| SPaRCNet (Jing et al., 2023) | 0.79M | 0.7896±0.0018 | 0.8414±0.0018 | 0.8676±0.0012 |
| ContraWR (Yang et al., 2023b) | 1.6M | 0.7746±0.0041 | 0.8421±0.0104 | 0.8456±0.0074 |
| CNN-Transformer (Peh et al., 2022) | 3.2M | 0.7777±0.0022 | 0.8433±0.0039 | 0.8461±0.0013 |
| FFCL (Li et al., 2022) | 2.4M | 0.7848±0.0038 | 0.8448±0.0065 | 0.8569±0.0051 |
| ST-Transformer (Song et al., 2021) | 3.5M | 0.7966±0.0023 | 0.8521±0.0026 | 0.8707±0.0019 |
| BIOT (Yang et al., 2023a) | 3.2M | 0.7959±0.0057 | 0.8792±0.0023 | 0.8815±0.0043 |
| LaBraM-Base | 5.8M | 0.8140±0.0019 | 0.8965±0.0016 | 0.9022±0.0009 |
| LaBraM-Large | 46M | 0.8226±0.0015 | 0.9130±0.0005 | 0.9127±0.0005 |
| LaBraM-Huge | 369M | **0.8258**±0.0011 | **0.9204**±0.0011 | **0.9162**±0.0016 |

Table 2: The results of different methods on TUEV.

| Methods | Model Size | Balanced Accuracy | Cohen's Kappa | Weighted F1 |
|---|---|---|---|---|
| SPaRCNet (Jing et al., 2023) | 0.79M | 0.4161±0.0262 | 0.4233±0.0181 | 0.7024±0.0104 |
| ContraWR (Yang et al., 2023b) | 1.6M | 0.4384±0.0349 | 0.3912±0.0237 | 0.6893±0.0136 |
| CNN-Transformer (Peh et al., 2022) | 3.2M | 0.4087±0.0161 | 0.3815±0.0134 | 0.6854±0.0293 |
| FFCL (Li et al., 2022) | 2.4M | 0.3979±0.0104 | 0.3732±0.0188 | 0.6783±0.0120 |
| ST-Transformer (Song et al., 2021) | 3.5M | 0.3984±0.0228 | 0.3765±0.0306 | 0.6823±0.0190 |
| BIOT (Yang et al., 2023a) | 3.2M | 0.5281±0.0225 | 0.5273±0.0249 | 0.7492±0.0082 |
| LaBraM-Base | 5.8M | 0.6409±0.0065 | 0.6637±0.0093 | 0.8312±0.0052 |
| LaBraM-Large | 46M | 0.6581±0.0156 | 0.6622±0.0136 | 0.8315±0.0040 |
| LaBraM-Huge | 369M | **0.6616**±0.0170 | **0.6745**±0.0195 | **0.8329**±0.0086 |

## 3.5 PRE-TRAINING WITH/WITHOUT DOWNSTREAM DATASETS

During the pre-training process, we hope that the model can learn general EEG representations that are not specific to any particular task. Although no label data is used during the pre-training process,

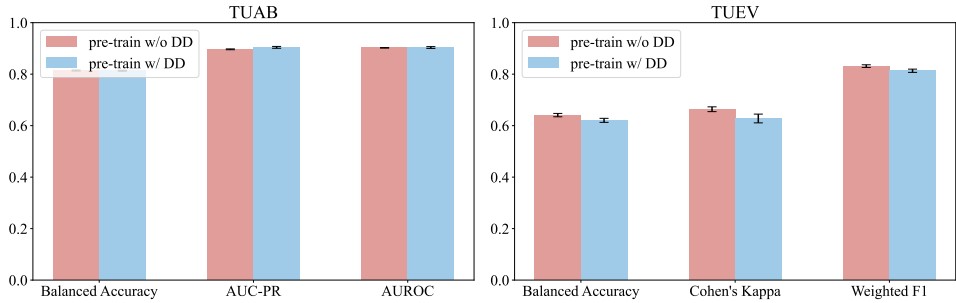

Figure 4: A comparison of the model's performance on the TUAB and TUEV datasets when incorporating themselves into the pre-training process or not.

to eliminate the influence of the pretraining data on downstream tasks, we compared the results with or without incorporating the downstream task dataset into the pre-training process or not. It is noted that the recordings of TUAB and TUEV are disjoint from recordings of pre-training datasets. As Figure 4 illustrates, the performance of the model on the downstream task was not significantly affected by whether or not to incorporate the downstream task datasets into the model's pre-training process. This demonstrates that our model has the capability to learn universal EEG representations, and provides guidance for the collection of more EEG data in the future. In other words, we do not need to expend a significant amount of effort on labeling EEG data during the pre-training process.

### 3.6 Scaling Data Size

Although we have collected approximately 2,500 hours of EEG data, it is still relatively small compared to the sample size in natural language processing and image processing. We answer **Q2** about the demand for data size to train LaBraMs with different sizes by scaling the pre-training data size. As illustrated in Figure 5, the performance of the Base model with 500 hours of training exceeds that of the 2500-hour model on TUAB, while approaching over 90% of the 2500-hour performance on TUEV. For the Large model, performance generally improves with increased data volume, though the growth rate slows after 1000 hours. In contrast, the Huge model exhibits a noticeable upward trend in performance as data size continues to expand. Therefore, we believe that with further expansion of the dataset, our model can achieve better performance. The question of how much EEG data is required for pre-training a large EEG model is undoubtedly an important issue worth exploring in this field. Nevertheless, 2,500 hours is not the answer to this question at least. Our observation basically follows the scaling law (Kaplan et al., 2020), from which we deduce that the Huge model would continue to perform better with the data size on the order of at least ten thousand hours.

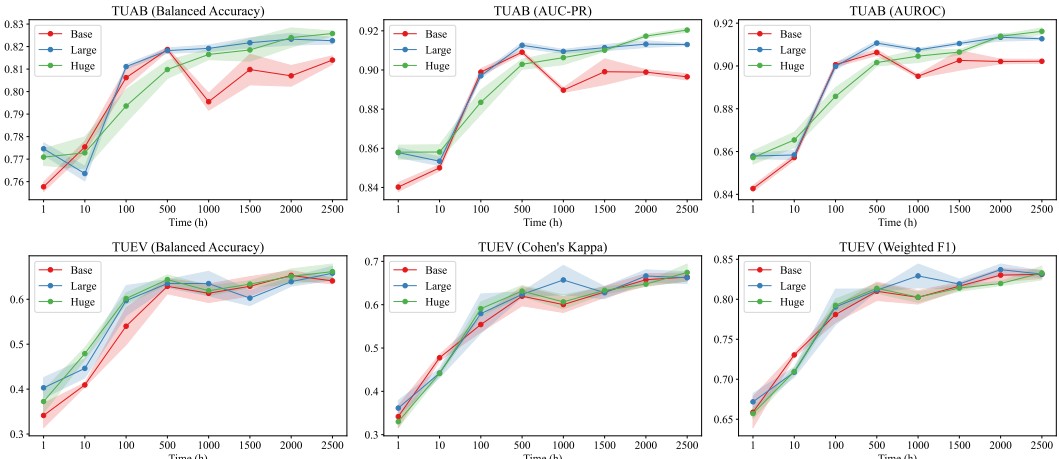

Figure 5: A comparison of the performance of the Base model, Large model, and Huge model on the TUAB and TUEV datasets as the pre-training data increases.

## 4 Conclusion

This paper proposes a Large Brain Model (LaBraM) that learns universal embeddings through unsupervised pre-training on over 2,500 hours of diverse EEG data. The LaBraM is capable of handling diverse EEG datasets due to the segmentation of raw EEG signals into channel patches and the use of vector-quantized neural spectrum prediction to generate a rich semantic tokenizer during pre-training. Additionally, the neural Transformer architecture enables effective representation learning of both temporal and spatial features of EEG signals, making it suitable for a wide range of downstream tasks in EEG analysis. The LaBraM was validated on multiple downstream tasks, including abnormal detection, event type classification, emotion recognition, and gait prediction. Our experiments show that the LaBraM outperforms all SOTA methods in their respective fields. In the end, we hope our work our can have implications for future developments in EEG-based deep learning models with improved perceptual capabilities and generalizability.

ACKNOWLEDGMENTS

This work was supported in part by grants from STI 2030-Major Projects+2022ZD0208500, Shanghai Municipal Science and Technology Major Project (Grant No. 2021SHZD ZX), Medical-Engineering Interdisciplinary Research Foundation of Shanghai Jiao Tong University "Jiao Tong Star" Program (YG2023ZD25), and GuangCi Professorship Program of RuiJin Hospital Shanghai Jiao Tong University School of Medicine.

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

## A  RELATED WORK

**Self-supervised Pre-training**. In recent years, self-supervised pre-training has made significant progress in natural language processing and computer vision. BERT (Devlin et al., 2018) innovatively proposed the idea of masking part of the input sentences and then reconstructing them. The GPT series (Radford et al., 2018; 2019; Brown et al., 2020) proposed to pre-train large language models by a large corpus of data in an autoregressive way. Both studies improved the fine-tuning performance significantly in various downstream tasks. In computer vision, iGPT (Chen et al., 2020) firstly brought the idea from GPT to pre-train a vision model. BEiT (Bao et al., 2022) pioneerly trained a vision tokenizer and leveraged BERT-like pre-training for training a vision Transformer. MAE (He et al., 2022) and SimMIM (Xie et al., 2022) practiced masked image modeling by simply reconstructing the raw pixels and achieved appreciable improvement.

**Learning with Heterogeneous Datasets.** MMM introduced a pre-training framework built on the unified topology and obtained topology-agnostic representations (Yi et al., 2023). Han et al. (2023) combined graph neural networks and transfer learning for non-invasive motor imagery EEG decoding with heterogeneous electrode configurations. Gu et al. (2023) developed two networks to learn from the shared and the complete channels across datasets, achieving coherent performance boosts. Liu et al. (2024) proposed a hierarchical personalized Federated Learning EEG decoding framework, enabling datasets with disparate data formats to collaborate in the model training process.

**Self-supervised Learning in BCI**. Although self-supervised pre-training has achieved great success, its potential in BCI is far from being explored. BENDR (Kostas et al., 2021) adapted Wav2vec 2.0 (Baevski et al., 2020), which uses contrastive learning to learn compressed representations of raw EEG signals. Banville *et al.* investigated temporal context prediction as well as contrastive predictive coding on two clinically relevant problems (Banville et al., 2021). ContraWR (Yang et al., 2023b), Contrast with the World Representation, used global statistics to distinguish signals associated with different sleep stages. BrainBERT (Wang et al., 2023) masks random parts of the stereo-electroencephalographic (SEEG) spectrogram and produce original embeddings with 43.6 hours of data. However, all existing studies either concentrate on specific BCI tasks or only employ small-size datasets and models, leaving room for exploring large-scale EEG data to train large EEG models through self-supervision.

## B  LaBraM PRE-TRAINING ANALYSIS

The pre-training of LaBraM can be interpreted as the training of a variational autoencoder (Kingma & Welling, 2014; Bao et al., 2022). We denote the original EEG sample as $x$, the corrupted EEG by masking as $x^{\mathcal{M}}$, and its Fourier spectrum (amplitude and phase) as $\tilde{x}$. The focus is on the evidence lower bound (ELBO) of the log-likelihood $p(\tilde{x}|x^{\mathcal{M}})$, which involves recovering the Fourier spectrum of the original image from the masked perspective:

$$\sum_{(x_i,x_i^{\mathcal{M}},\tilde{x}_i)\in\mathcal{D}} \log p(\tilde{x}_i|x_i^{\mathcal{M}}) \geq \sum_{(x_i,x_i^{\mathcal{M}},\tilde{x}_i)\in\mathcal{D}} (\mathbb{E}_{z_i\sim q_\phi(\boldsymbol{z}|x_i)}(\log p_\psi(\tilde{x}_i|z_i) - D_{KL}(q_\phi(\boldsymbol{z}|x_i), p_\theta(\boldsymbol{z}|x_i^{\mathcal{M}})),$$

(13)

where $q_\phi(\boldsymbol{z}|x)$ represents the neural tokenizer that encodes the EEG sample into discrete neural tokens, $p_\psi(\tilde{x}|z)$ denotes the neural decoder predicting the Fourier spectrum from given neural tokens, and $p_\theta(\boldsymbol{z}|x^{\mathcal{M}})$ is the LaBraM pre-training for masked EEG modeling, where the LaBraM encoder reconstructs neural tokens from the corrupted EEG input.

The whole framework is optimized through a two-stage procedure as (Van Den Oord et al., 2017). For the first stage, we train the neural tokenizer as a discrete variational autoencoder by minimizing the reconstruction loss $-\mathbb{E}_{z_i\sim q_\phi(\boldsymbol{z}|x_i)}(\log p_\psi(\tilde{x}_i|z_i)$ with a uniform prior. For the second stage, we set $q_\phi$ as well as $p_\psi$ fixed and learn the prior $p_\theta$ by minimizing the loss $D_{KL}$. For simplicity, $q_\phi(\boldsymbol{z}|x_i)$ is defined as a one-point distribution with the most likely neural tokens $\hat{z}_i = \arg\max_z q_\phi(\boldsymbol{z}|x_i)$. Consequently, we can rewrite Equation 13 as

$$\sum_{(x_i,x_i^{\mathcal{M}},\tilde{x}_i)\in\mathcal{D}} (\mathbb{E}_{z_i\sim q_\phi(\boldsymbol{z}|x_i)}(\log p_\psi(\tilde{x}_i|z_i) + \log p_\theta(\hat{z}_i|x_i^{\mathcal{M}})),$$

(14)

where the first term is the objective for vector-quantized neural spectrum prediction and the second term is the objective for LaBraM pre-training.

# C  HYPERPARAMETER SETTINGS

Table 3: Hyperparameters for vector-quantized neural spectrum prediction training.

| Hyperparameters | | Values |
|---|---|---|
| Temporal Encoder | Iput channels | {1,8,8} |
| | Output channels | {8,8,8} |
| | Kernel size | {15,3,3} |
| | Stride | {8,1,1} |
| | Padding | {7,1,1} |
| Transformer encoder layers | | 12 |
| Transformer decoder layers | | 3 |
| Hidden size | | 200 |
| MLP size | | 800 |
| Attention head number | | 10 |
| Codebook size | | 8192×64 |
| Batch size | | 1024 |
| Peak learning rate | | 5e-5 |
| Minimal learning rate | | 1e-5 |
| Learning rate scheduler | | Cosine |
| Optimizer | | AdamW |
| Adam $\beta$ | | (0.9,0.99) |
| Weight decay | | 1e-4 |
| Total epochs | | 100 |
| Warmup epochs | | 10 |
| Data stride | | 200 |

Table 4: Hyperparameters for masked EEG pre-training.

| Hyperparameters | | LaBraM-Base | LaBraM-Large | LaBraM-Huge |
|---|---|---|---|---|
| Temporal Encoder | Iput channels | {1,8,8} | {1,16,16} | {1,32,32} |
| | Output channels | {8,8,8} | {16,16,16} | {32,32,32} |
| | Kernel size | | {15,3,3} | |
| | Stride | | {8,1,1} | |
| | Padding | | {7,1,1} | |
| Transformer encoder layers | | 12 | 24 | 48 |
| Hidden size | | 200 | 400 | 800 |
| MLP size | | 800 | 1600 | 3200 |
| Attention head number | | 10 | 16 | 16 |
| Batch size | | | 512 | |
| Peak learning rate | | | 5e-4 | |
| Minimal learning rate | | | 1e-5 | |
| Learning rate scheduler | | | Cosine | |
| Optimizer | | | AdamW | |
| Adam $\beta$ | | | (0.9,0.98) | |
| Weight decay | | | 0.05 | |
| Total epochs | | | 50 | |
| Warmup epochs | | | 5 | |
| Data stride | | | 800 | |
| Gradient clipping | | | 3 | |
| Layer scale init | | 0.1 | 1e-5 | 1e-6 |
| EMA weight | | | 0.996 | |
| Mask ratio | | | 0.5 | |

Table 5: Hyperparameters for downstream fine-tuning.

| Hyperparameters | Values |
|---|---|
| Batch size | 512 |
| Peak learning rate | 5e-4 |
| Minimal learning rate | 1e-6 |
| Learning rate scheduler | Cosine |
| Optimizer | AdamW |
| Adam $\beta$ | (0.9,0.999) |
| Weight decay | 0.05 |
| Total epochs | 50 (B) 30 (L/H) |
| Warmup epochs | 5 (B) 3 (L/H) |
| Drop path | 0.1 (B/L) 0.2 (H) |
| Layer-wise learning rate decay | 0.65 (B) 0.8 (L/H) |
| Label smoothing (multi-class classification) | 0.1 |

## D  PRE-TRAINING DATASET DESCRIPTION

We describe the datasets we use for training LaBraM here.

Training datasets (for both vector-quantized neural spectrum prediction training and LaBraM pre-training, the total time is 2534.78 hours):

- **BCI Competition IV-1** (Blankertz et al., 2007): A motor imagery dataset containing 59 EEG channels at 1000Hz sampling rate for 2 classes of left hand, right hand, foot (+ idle state) for 7 subjects. The recording was made using BrainAmp MR plus amplifiers and an Ag/AgCl electrode cap. (total time: 8.21 hours)

- **Emobrain** (Savran et al., 2006): A multimodal emotion dataset where EEG (64 channels, 1024 Hz) and fNIRS, are recorded by the Biosemi Active 2 acquisition system, including 16 subjects. The emotions were elicited through a selected subset of IAPS dataset. (total time: 4.94 hours)

- **Grasp and Lift EEG Challenge** (Luciw et al., 2014): A dataset containing EEG recordings (32 channels, 500 Hz) of 12 subjects performing grasp-and-lift (GAL) trials. The EEG cap was used in conjunction with a BrainAmp EEG signal amplifier. (total time: 11.72 hours)

- **Inria BCI Challenge** (Margaux et al., 2012): A P300-based spelling dataset including 26 subjects with EEG records (56 channels, 600 Hz) by Ag/AgCl EEG sensors (VSM-CTF compatible system). (total time: 29.98 hours)

- **EEG Motor Movement/Imagery Dataset** (Schalk et al., 2004): A motor imagery dataset consisting of 109 volunteers performing 2 baseline tasks (eye-open and eye-closed), motor movement, and motor imagery (both fists or both feet) with EEG records (64 channels, 160 Hz) using the BCI2000 system. (total time: 47.3 hours)

- **Raw EEG Data** (Trujillo, 2020): A dataset where EEG (64 channels, 256 Hz) was recorded during the reported Information-Integration categorization task and the reported multidimensional Rule-Based categorization task. (total time: 34.35 hours)

- **Resting State EEG Data** (Trujillo et al., 2017): A dataset comprising 22 subjects for a resting task of 8 mins with 4 mins of eyes closed and 4 mins of eyes open with 64 EEG channels at 256 Hz using active Ag/AgCl electrodes either mounted in a BioSemi electrode cap or via freestanding electrodes. (total time: 3.04 hours)

- **SEED Series** (Zheng & Lu, 2015; Zheng et al., 2018; Liu et al., 2022): The emotional datasets including SEED (15 subjects), SEED-IV (15 subjects), SEED-GER (8 subjects), and SEED-FRA (8 subjects). All EEG signals (62 channels, 1000 Hz) were recorded with the ESI NeuroScan System in response to videos. (total time: 166.75 hours)

- **Siena Scalp EEG Database** (Detti et al., 2020): A database consisting of EEG recordings (31 channels, 512 Hz) of 14 patients employing EB Neuro and Natus Quantum LTM amplifiers, and reusable silver/gold cup electrodes. (total time: 30.47 hours)

- **SPIS Resting State Dataset** (Torkamani-Azar et al., 2020): A dataset including 10 subjects, 2.5 minutes recording in each state (eyes-closed and eyes-open) prior to a 105-minute session of Sustained Attention to Response Task with fixed-sequence and varying ISIs. Monopolar EEG activity (64 channels, 2048 Hz) was collected via 64 Ag/AgCl active electrodes. (total time: 0.83 hour)

- **Target Versus Non-Target** (Korczowski et al., 2019): A dataset including 50 subjects playing Brain Invaders, a visual P300 Brain-Computer Interface using oddball paradigm with adapative Riemannian Geometry (no-calibration). EEG signals (32 channels, 512 Hz) were acquired by means of a research-grade amplifier (g.USBamp, g.tec, Schiedlberg, Austria) and the g.GAMMAcap. (total time: 16 hours)

- **TUAR** (Buckwalter et al., 2021): This subset of TUEG contains annotations of 5 different artifacts with EEG recorded (23 channels, 256 Hz). (total time: 92.22 hours)

- **TUEP** (Veloso et al., 2017): This is a subset of TUEG that contains 100 subjects with epilepsy and 100 subjects without epilepsy with EEG recorded (19-23 channels, 256 Hz), as determined by a certified neurologist. (total time: 591.22 hours)

- **TUSZ** (Shah et al., 2018): This corpus has EEG signals that have been manually annotated data for seizure events (start time, stop, channel, and seizure type) with EEG recorded (19-23 channels, 256 Hz). (total time: 1138.53 hours)

- **TUSL** (von Weltin et al., 2017): This is another subset of TUEG that contains annotations of slowing events (23 channels, 256 Hz). This corpus has been used to study common error modalities in automated seizure detection. (total time: 20.59 hours)

- **Self-collected EEG Data** (Jiang et al., 2023; 2021; Luo et al., 2022; Li et al., 2021; Tao & Lu, 2020): We further collect EEG data from more than 140 subjects by ourselves (62 channels, 1000 Hz) with the ESI NeuroScan System. (total time: 342.23 hours)

# E  VISUALIZATION OF VECTOR-QUANTIZED NEURAL SPECTRUM PREDICTION

We further visualize how the amplitude and phase in the Fourier domain are reconstructed. As depicted in Figure 7, although some details are missing, the overall trend of the amplitude is reconstructed well. In contrast, the reconstruction of the phase is not as good as the amplitude. Nevertheless, it can be seen from Figure 6 that there is still a stable decrease in the reconstruction loss during training, which indicates the discrete codebook does learn high-level information from the Fourier domain.

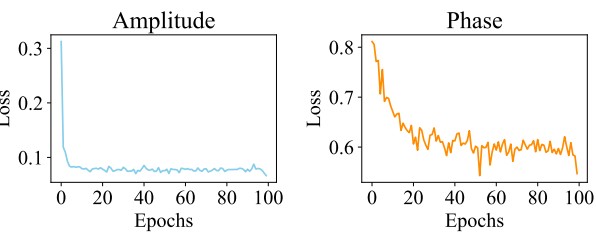

Figure 6: The reconstruction loss curve of amplitude and phase.

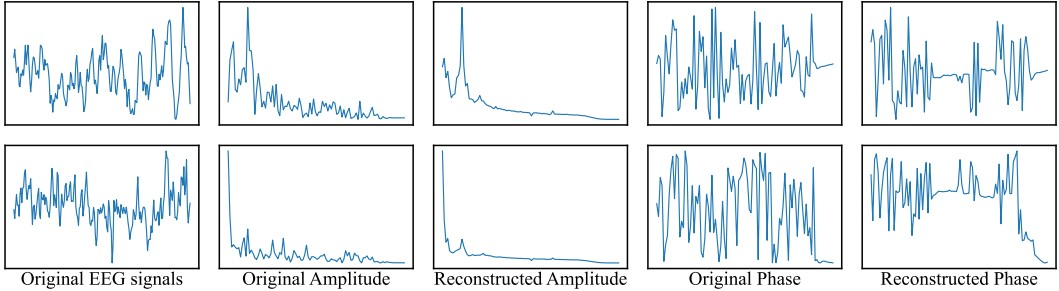

Figure 7: Visualization of reconstructed Fourier spectrum. Note that we only visualize half of the results since DFT is conjugate symmetric.

# F  MORE EXPERIMENTS ON OTHER BCI TASKS

We conduct two additional BCI tasks on the following datasets:

- **SEED-V** (emotion recognition) (Liu et al., 2021): An emotion EEG dataset containing five emotion categories (happy, sad, neutral, disgust, and fear). The experiment collected EEG data (62 channels, 1000 Hz) from 20 subjects, including 10 males and 10 females. Each subject participated in the experiments three times and each session included fifteen video clips corresponding to the five emotions, where each video clip lasted for several minutes. The EEG signals are segmented into 148,080 1-second samples.
- **MoBI** (gait prediction) (He et al., 2018): A mobile brain-body imaging dataset acquired during treadmill walking in a BCI task, which is a lower limb motor imagery dataset. Six goniometers were employed to record bilateral joint angles on the legs (hip, knee, and ankle). The objective is to regress the angles for 12 targets (left leg and right leg). The data were collected from 8 healthy subjects, each of whom had three identical trials. The EEG signals (60 channels, 100 Hz) were recorded by the ActiCap system. Setting the stride to 50 ms, the dataset involves 575,830 2-second samples.

For SEED-V, as there are fifteen trials for one session, we separate the fifteen trials into three parts with an equal number of trials, i.e., 5:5:5. We merge each part from all sessions of subjects and derive the training, validation, and test set. As SEED-V is overall balanced, we consider accuracy instead of balanced accuracy as a metric to compare performance. Note that some implementation details are a bit different from default settings on this dataset due to different characteristics (peak learning rate: 5e-4 (L) 5e-3 (H); total epochs: 50 (L/H), warmup epochs: 4 (L) 5 (H)).

For MoBI, each trial consisted of a 15-minute treadmill walking session (training session), followed by a 5-minute treadmill walking session (test session) with a closed-loop BCI. To validate the model, we split the training session into two parts: the first 10 minutes of EEG and its corresponding joint data were used as training data, while the last 5 minutes of data were used as validation data. Meanwhile, we combined all the training data, validation data, and testing data of the eight subjects to form corresponding larger training datasets, validation datasets, and testing datasets. Since most angles are typically lower than $90°$, the target angles are divided by 90 for normalization. We report the average value of 12 targets for each metric.

As the task of MoBI is regression, we choose the following metrics to evaluate the performance of different methods: 1) **Pearson's correlation**: Pearson's correlation coefficient which is used to quantify the models' regression effect. It measures the linear correlation between two variables X and Y. 2) **R2 score**: $R^2$ (coefficient of determination) regression score function, which measures how well a statistical model predicts an outcome. 3) **RMSE**: Root Mean Square Error is the standard deviation of the residuals (prediction errors). R2 score is utilized as the monitor to select the best model. MSE loss is the objective to optimize the models.

The experimental results are presented in Figure 6. On SEED-V, LaBraMs outperform all baseline methods on all metrics. The phenomenon that the performance increases when the model gets larger is also observed. For MoBI, our Base model archives competitive results compared to the best baseline method. Whereas, the Large and Huge models obtain better performance among all methods.

Table 6: The results of different methods on SEED-V and MoBI.

| | SEED-V | | | MoBI | | |
|---|---|---|---|---|---|---|
| | Accuracy | Cohen's Kappa | Weighted F1 | Pearson's Correlation | R2 Score | RMSE↓ |
| SPaRCNet | 0.2887±0.0047 | 0.1032±0.0083 | 0.2904±0.0064 | 0.4561±0.0161 | 0.1467±0.0064 | 0.1344±0.0006 |
| ContraWR | 0.3603±0.0098 | 0.1988±0.0114 | 0.3590±0.0091 | 0.3357±0.0164 | 0.0743±0.0093 | 0.1401±0.0008 |
| CNN-Transformer | 0.3665±0.0058 | 0.2034±0.0060 | 0.3638±0.0065 | 0.3224±0.0109 | 0.0628±0.0089 | 0.1411±0.0007 |
| FFCL | 0.3686±0.0059 | 0.2094±0.0078 | 0.3679±0.0062 | 0.3158±0.0235 | 0.0712±0.0124 | 0.1396±0.0014 |
| ST-Transformer | 0.2772±0.0047 | 0.0783±0.0071 | 0.2625±0.0061 | 0.5442±**0.0012** | 0.2911±**0.0014** | 0.1222±**0.0001** |
| BIOT | 0.3802±0.0094 | 0.2247±0.0100 | 0.3809±0.0114 | 0.2757±0.0173 | 0.0597±0.0069 | 0.1401±0.0006 |
| LaBraM-Base | 0.4095±0.0062 | 0.2613±0.0075 | 0.4120±0.0057 | 0.5383±0.0102 | 0.2876±0.0032 | 0.1225±0.0003 |
| LaBraM-Large | 0.4096±0.0075 | 0.2639±0.0090 | 0.4127±0.0079 | 0.5603±0.0020 | 0.3093±0.0032 | 0.1197±0.0003 |
| LaBraM-Huge | **0.4102**±**0.0037** | **0.2646**±**0.0046** | **0.4136**±**0.0047** | **0.5632**±0.0023 | **0.3145**±0.0032 | **0.1196**±0.0003 |

# G    EFFECTIVENESS OF VECTOR-QUANTIZED NEURAL SPECTRUM PREDICTION

To verify the effectiveness of vector-quantized neural spectrum prediction, we elaborate on three types of experimental settings as illustrated in Table 7. The comparison between LaBraM and Setting 1 demonstrates that the codebook is effective for masked EEG modeling. LaBraM obtains the best performance on TUEV and the lowest standard deviations on TUAB. There is an interesting observation that masked EEG modeling with the assistance of training an auxiliary neural tokenizer (LaBraM and Setting 1) performs greatly better on TUEV while the naive masked EEG modeling (Setting 2 and Setting 3) performs slightly better on TUAB. One explanation for this phenomenon is that learning semantic representations from the neural tokenizer and codebook significantly benefits high-level downstream tasks like TUEV which classifies different types of events. Whereas, TUAB is a low-level downstream task where the clinically normal/abnormal EEG segments can be easily distinguished visually. Hence, simply reconstructing origin signals or their Fourier spectrum is able to perform well on these low-level tasks but fails to obtain satisfying performance on high-level tasks.

Table 7: Ablations to validate the effectiveness of vector-quantized neural spectrum prediction.

| | TUAB | | | TUEV | | |
|---|---|---|---|---|---|---|
| | Balanced Accuracy | AUC-PR | AUROC | Balanced Accuracy | Cohen's Kappa | Weighted F1 |
| LaBraM | 0.8140±**0.0019** | 0.8965±**0.0016** | 0.9022±**0.0009** | **0.6409**±0.0065 | **0.6637**±0.0093 | **0.8312**±0.0052 |
| Setting 1 | 0.8058±0.0044 | 0.8949±0.0037 | 0.8964±0.0012 | 0.6162±0.0174 | 0.6376±0.0168 | 0.8170±0.0058 |
| Setting 2 | **0.8261**±0.0030 | **0.9150**±0.0016 | **0.9067**±0.0024 | 0.5630±0.0313 | 0.5910±0.0156 | 0.7979±0.0082 |
| Setting 3 | 0.8166±0.0073 | 0.9062±0.0029 | 0.9053±0.0026 | 0.5730±0.0133 | 0.5643±**0.0089** | 0.7819±**0.0040** |

Setting 1: We directly predict output embeddings of the neural tokenizer by maximizing cosine similarity instead of predicting the discrete neural tokens from the codebook.
Setting 2: We discard the neural tokenizer and directly reconstruct raw EEG patches by minimizing MSE loss.
Setting 3: We discard the neural tokenizer and reconstruct the Fourier spectrum (amplitude and phase) of raw EEG patches by minimizing MSE loss.

# H    ABLATION ON MASK RATIO

In this experiment, we conduct different settings of the mask ratio to explore its impact. It is noted that we introduce the symmetric masking strategy, so we only need to validate half of the mask ratios. As the mask ratio is set to $r$, the symmetric masking will mask $1 - r$ proportion of EEG patches. The ablation results are provided in Table 8, where experiments are conducted on TUAB and TUEV. It can be induced that the best mask ratio is 0.4 (0.6) for TUAB and 0.5 (0.5) for TUEV. Moreover, 0.5 (0.5) is the second-best mask ratio for TUAB while the remaining mask ratios are incredibly close. The performance for mask ratios except 0.5 (0.5) is also similar to each other. Notably, the mask ratio of 0.5 (0.5) achieves smaller standard deviations on both TUAB and TUEV. Therefore, we conclude that 0.5 (0.5) is a relatively good mask ratio for the masked EEG modeling of LaBraM pre-training.

Table 8: Performance of different mask ratios.

| Mask Ratio | TUAB | | | TUEV | | |
|---|---|---|---|---|---|---|
| | Balanced Accuracy | AUC-PR | AUROC | Balanced Accuracy | Cohen's Kappa | Weighted F1 |
| 0.5 (0.5) | 0.8140±**0.0019** | 0.8965±0.0016 | 0.9022±0.0009 | **0.6409**±0.0065 | **0.6637**±0.0093 | **0.8312**±0.0052 |
| 0.4 (0.6) | **0.8145**±0.0039 | **0.9083**±0.0030 | **0.9049**±0.0038 | 0.6174±0.0127 | 0.6123±0.0094 | 0.8067±0.0059 |
| 0.3 (0.7) | 0.7994±0.0037 | 0.8950±**0.0006** | 0.8974±0.0008 | 0.6112±0.0216 | 0.6089±0.0158 | 0.8068±0.0086 |
| 0.2 (0.8) | 0.8039±0.0054 | 0.8990±0.0050 | 0.9018±0.0023 | 0.6054±0.0268 | 0.6050±0.0152 | 0.8024±0.0089 |
| 0.1 (0.9) | 0.8022±0.0041 | 0.8968±0.0010 | 0.8992±**0.0007** | 0.6033±0.0264 | 0.6181±0.0178 | 0.8134±0.0094 |

## I    ABLATION ON SYMMETRIC MASKING

We conduct an ablation study to verify the contribution of the symmetric masking strategy. Table 9 reports the results on TUAB and TUEV. It is obvious that the performance of most metrics decreases by a remarkable margin on both datasets, especially TUEV. Specifically, without symmetric masking, the performance of the base model increases a little bit on TUAB. Nevertheless, the performance decreases in most other scenarios. This is because the data is sufficient for the base model, so the symmetric masking strategy which acts like data augmentation contributes a little to the model training. For larger models like LaBraM-Huge, the symmetric masking improves the downstream performance as it requires more data. This observation indicates that symmetric masking can not only boost the downstream performance but also improve stability and robustness.

Table 9: Ablation study of symmetric masking (SM).

| | TUAB | | | TUEV | | |
|---|---|---|---|---|---|---|
| | Balanced Accuracy | AUC-PR | AUROC | Balanced Accuracy | Cohen's Kappa | Weighted F1 |
| LaBraM-Base | 0.8140±**0.0019** | 0.8965±**0.0016** | 0.9022±**0.0009** | **0.6409**±**0.0065** | **0.6637**±**0.0093** | **0.8312**±**0.0052** |
| w/o SM | **0.8155**±0.0041 | **0.9077**±0.0069 | **0.9065**±0.0034 | 0.6284±0.0175 | 0.6279±0.0260 | 0.8152±0.0105 |
| LaBraM-Large | **0.8226**±**0.0015** | 0.9130±**0.0005** | **0.9127**±**0.0005** | **0.6581**±0.0156 | **0.6622**±0.0136 | **0.8315**±0.0040 |
| w/o SM | 0.8198±0.0042 | **0.9140**±0.0007 | 0.9106±0.0012 | 0.6548±0.0246 | 0.6601±**0.0122** | 0.8319±**0.0034** |
| LaBraM-Huge | **0.8258**±0.0011 | **0.9204**±0.0011 | **0.9162**±0.0016 | **0.6616**±0.0170 | **0.6745**±0.0195 | **0.8329**±0.0086 |
| w/o SM | 0.8247±**0.0010** | 0.9188±**0.0005** | 0.9149±**0.0004** | 0.6261±0.0178 | 0.6391±**0.0179** | 0.8152±**0.0085** |

## J    LaBraM WITHOUT PRE-TRAINING

In this experiment, we directly train LaBraM on the downstream datasets from scratch without pre-training to validate the effectiveness of the masked EEG modeling pre-training. The steep performance drop demonstrates the usefulness of pre-training, as illustrated in Figure 8.

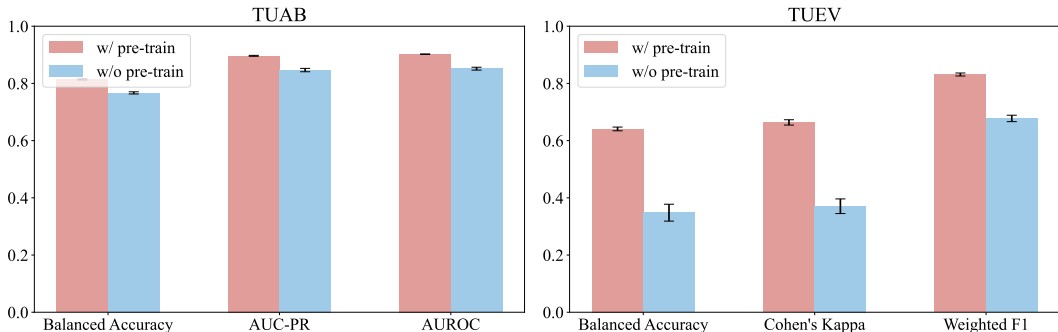

Figure 8: Comparison with model without pre-training.

## K    PARTIAL FINE-TUNING

In Table 10, we report the results about fine-tuning part of LaBraM. We elaborate on several settings: fine-tuning all 12 Transformer blocks, fine-tuning the last 8 Transformer blocks, fine-tuning the last 4 Transformer blocks, and linear probing. It is noteworthy that for linear probing, we set the weight decay to 0. One can see that on TUAB, the results of full fine-tuning, fine-tuning 12 Transformer blocks, and fine-tuning 8 Transformer blocks are quite similar. When only fine-tuning 4 Transformer blocks and linear probing, there is a slight degradation in performance. On TUEV, yet, fine-tuning 8 Transformer blocks achieves the best performance on all three metrics. Notably, the results of linear probing are much worse than other settings, which still have room for improvement.

Table 10: Results of fine-tuning part of LaBraM.

| Fine-tuning Part | TUAB | | | TUEV | | |
|---|---|---|---|---|---|---|
| | Balanced Accuracy | AUC-PR | AUROC | Balanced Accuracy | Cohen's Kappa | Weighted F1 |
| All | 0.8140±**0.0019** | **0.8965**±0.0016 | **0.9022**±**0.0009** | 0.6409±**0.0065** | 0.6637±0.0093 | 0.8312±0.0052 |
| Transformer (12) | **0.8141**±0.0022 | 0.8963±**0.0014** | **0.9022**±**0.0009** | 0.6541±0.0250 | 0.6782±0.0189 | 0.8386±0.0090 |
| Transformer (8) | 0.8134±0.0022 | 0.8960±0.0019 | 0.9020±**0.0009** | **0.6611**±0.0152 | **0.6820**±**0.0089** | **0.8406**±**0.0036** |
| Transformer (4) | 0.8074±0.0032 | 0.8930±0.0065 | 0.8967±0.0018 | 0.6188±0.0118 | 0.6560±0.0233 | 0.8256±0.0114 |
| Linear Probe | 0.7954±0.0059 | 0.8864±0.0030 | 0.8835±0.0028 | 0.3461±0.0225 | 0.3968±0.0329 | 0.6974±0.0161 |

## L  ABLATION ON SPATIAL EMBEDDINGS

The spatial embeddings have helped us address the challenge of heterogeneity in electrode configurations. However, it is important to verify the effectiveness of this approach. During pre-training, we observed that the loss could not converge without spatial embeddings. This was expected, as the model needs spatial embeddings to identify the masked patch to reconstruct. During fine-tuning on downstream datasets, we discard the spatial embeddings and notice a significant drop in performance, as shown in Table 11. This clearly demonstrates the importance of spatial embeddings in capturing spatial information.

Table 11: Ablation study of spatial embeddings (SE).

| | TUAB | | | TUEV | | |
|---|---|---|---|---|---|---|
| | Balanced Accuracy | AUC-PR | AUROC | Balanced Accuracy | Cohen's Kappa | Weighted F1 |
| LaBraM | **0.8140**±**0.0019** | **0.8965**±**0.0016** | **0.9022**±**0.0009** | **0.6409**±**0.0065** | **0.6637**±**0.0093** | **0.8312**±**0.0052** |
| w/o SE | 0.8004±0.0037 | 0.8922±0.0023 | 0.8888±0.0018 | 0.5949±0.0423 | 0.6069±0.0248 | 0.8040±0.0111 |

## M  DISCUSSION

**Limitations**. First of all, although we have collected the largest EEG dataset ever of over 2,500 hours and trained the largest model with 369M parameters ever for BCI, it still has a large margin from today's large vision models and large language models. Our work is only the first step to explore the feasibility of training a large EEG model for learning generic representations. It is delighted to find that training a large EEG model with tremendous EEG data does work and obtain appreciable performance gain compared to existing methods developed for specific BCI tasks. Secondly, LaBraM needs to be fully fine-tuned to adapt to downstream tasks, which might be computation-costly and memory-costly. Finally, LaBraM is trained with unimodal EEG data. It is worthwhile to investigate training large EEG models with other modalities.

**Outlook**. In view of the above limitations, our paradigm paves the way for further research, encompassing the following aspects: 1) Collecting more EEG data from a variety of BCI tasks, and training a larger EEG model to see whether emergent abilities exist in the EEG model similar to large language models; 2) Leveraging the parameter efficient learning methods, such as adapters, prompt tuning, and LoRA, to reduce the fine-tuning overhead and save space for disks; 3) Incorporating other modalities like image, language, speech, and other physiological signals into large EEG models training to build new paradigms, or aligning EEG representations with other modalities in semantic space, which can be a meaningful and challenging direction for future work.

