# OpenReview forum: "Large Brain Model for Learning Generic Representations with Tremendous EEG Data in BCI"
_ICLR.cc/2024/Conference — ICLR 2024 spotlight_

### Official Review · Reviewer_ztTZ · 2023-10-31

**Soundness:** 3 good
**Presentation:** 4 excellent
**Contribution:** 4 excellent
**Rating:** 8
**Confidence:** 2

**Summary:**

This manuscript introduces the Large Brain Model (LaBraM), an innovative approach designed to enhance the capabilities and scalability of EEG-based deep learning models in brain-computer interaction (BCI). The authors explore the potential of Large EEG Models (LEMs) for universal perceptual capabilities through unsupervised pre-training, followed by fine-tuning for various downstream tasks. However, EEG data presents unique challenges like mismatched electrodes, diverse data lengths, and low signal-to-noise ratios. To address these, LaBraM implements cross-dataset learning by segmenting EEG signals into channel patches and employs vector-quantized neural spectrum prediction for rich neural tokenization. This is further enhanced by pre-training neural Transformers to predict original neural codes for masked EEG channel patches. LaBraM is presented in three sizes, Base (5.8M), large (46M), and Huge (369M), with the "Huge" variant being the biggest EEG model to date. Pre-trained on approximately 2,500 hours of EEG signals from around 20 datasets, LaBraM showcases superior performance across several downstream tasks, outshining existing state-of-the-art methods. The authors also delve into the data requirements for training various sizes of the model.

**Strengths:**

1. The manuscript represents a significant contribution to the field, pioneering the development of the largest Large EEG Model (LEM) for EEG decoding. By effectively addressing two pivotal challenges in this field — the utilization of large-scale EEG data and the required data volume — this work lays a solid foundation for future researchers in this field
2. The clarity and coherence of the presentation are commendable, facilitating an in-depth understanding of the proposed methodologies.
3. A notable strength of this work is the comprehensive experimental evaluation. The authors have conducted a plethora of experiments, providing supplementary results, detailed ablation studies, and a thorough discussion on hyperparameter settings in the appendix.
4. The Figures, complemented by lucid annotations, further enhance the comprehensibility and accessibility of the work to the audience.

**Weaknesses:**

1. While the authors acknowledge the challenge of varying configurations in EEG data collection, particularly concerning electrode variations across different datasets, the manuscript does not provide a comprehensive solution to this challenge. Specifically, the Temporal & Spatial embedding section offers an ambiguous explanation regarding spatial embedding (SE). The method appears to encode channels based merely on their sequential order, which may not accurately represent the channel's functional significance or location in the brain. Given the heterogeneity in electrode configurations and positions across various datasets, it's imperative for the authors to elucidate how their approach effectively manages this inconsistency, and use qualitative results to demonstrate the model has learned different electrode configurations.

2. The paper draws parallels to another LEM, BIOT, developed by Yang et al., and even follows some experimental settings from the same. Given the apparent similarities, it remains unclear as to what drives the performance improvements of the proposed model — is it solely attributed to the increased model size, enhanced training data volume, or specific architectural designs? A more in-depth comparative discussion and analysis between the two models would be beneficial to ascertain the genuine contributions and innovations of the current work.

**Questions:**

Will the pre-trained checkpoints be released for open-source development as well?

---

> ### Author Response · Authors · 2023-11-17
>
> We are deeply grateful to you for the thorough review and constructive comments, which have greatly assisted us in improving the quality and presentation of our manuscript. Please see below for the point-by-point responses to your comments.
>
> > [**W1**] While the authors acknowledge the challenge of varying configurations in EEG data collection, particularly concerning electrode variations across different datasets, the manuscript does not provide a comprehensive solution to this challenge. Specifically, the Temporal & Spatial embedding section offers an ambiguous explanation regarding spatial embedding (SE). The method appears to encode channels based merely on their sequential order, which may not accurately represent the channel's functional significance or location in the brain. Given the heterogeneity in electrode configurations and positions across various datasets, it's imperative for the authors to elucidate how their approach effectively manages this inconsistency, and use qualitative results to demonstrate the model has learned different electrode configurations.
>
> * Thanks for your valuable suggestion. First of all, our spatial embeddings are initialized according to the international 10-20 system (https://en.wikipedia.org/wiki/10%E2%80%9320_system_(EEG)) which was commonly followed by most existing EEG datasets. Though different datasets may have different configurations, they all utilize a subset of electrodes of the international 10-20 system, providing the possibility to solve this problem. Specifically, our solution is adding the corresponding spatial embedding according to the international 10-20 system. For example, both dataset A and dataset B have the electrode FP1, so the FP1 electrodes in the two datasets are added by the same spatial embedding. By this means, the spatial embeddings incorporate the spatial information into the model training, which is unremovable during pre-training. If we try to discard the spatial embeddings, the pre-training loss cannot converge because it does not have the ability to identify which masked electrodes to reconstruct. To demonstrate the effectiveness of the spatial embeddings, we provide the ablation study where the spatial embeddings are directly discarded during fine-tuning on downstream datasets as follows:
>
> | TUAB | Balanced Accuracy | AUC-PR | AUROC |
> | -------------------- | :---------------------: | :--------------------: | :------------------: |
> | LaBraM | 0.8140 $\pm$ 0.0019 | 0.8965 $\pm$ 0.0016 | 0.9022 $\pm$ 0.0009 |
> | w/o spatial embedding | 0.8004 $\pm$ 0.0037 | 0.8922 $\pm$ 0.0023 | 0.8888 $\pm$ 0.0018 |
>
> |TUEV|Balanced Accuracy|Cohen's Kappa|Weighted F1|
> |---|---|---|---|
> |LaBraM|0.6409 $\pm$ 0.0065|0.6637 $\pm$ 0.0093|0.8312 $\pm$ 0.0052|
> |w/o spatial embedding|0.5949 $\pm$ 0.0423|0.6069 $\pm$ 0.0248|0.8040 $\pm$ 0.0111|
>
> The results clearly demonstrate the importance of spatial embeddings in capturing spatial information. We have updated this experiment to the revised paper in Appendix L.
>
> > [**W2**] The paper draws parallels to another LEM, BIOT, developed by Yang et al., and even follows some experimental settings from the same. Given the apparent similarities, it remains unclear as to what drives the performance improvements of the proposed model — is it solely attributed to the increased model size, enhanced training data volume, or specific architectural designs? A more in-depth comparative discussion and analysis between the two models would be beneficial to ascertain the genuine contributions and innovations of the current work.
>
> * Thanks for your valuable feedback. Our conclusion to your concern is that the well-designed large-scale self-supervised pre-training contributes most to our superior performance compared to BIOT. In Appendix J, we have conducted ablation to see whether the pre-training really works. It can be seen that without pre-training, the performance decreases steeply on the two datasets especially on TUEV.
>
> > [**Q1**] Will the pre-trained checkpoints be released for open-source development as well?
>
> * Yes, we will release the pre-trained models as well as the codes.

---

> > ### Comment · Reviewer_ztTZ · 2023-11-23
> >
> > Thank you for the thorough response to my concerns and for incorporating additional results. Your efforts in conducting further experiments are greatly appreciated.

---

### Official Review · Reviewer_WxGC · 2023-10-31

**Soundness:** 3 good
**Presentation:** 4 excellent
**Contribution:** 3 good
**Rating:** 8
**Confidence:** 4

**Summary:**

The authors introduce a Transformer-based model and a pretraining methodology to learn representations from EEG data in a self-supervised manner on a large collection of datasets. The approach combines two parts. First, a vector-quantized tokenizer (similar to a VQ-VAE) is trained to quantize 1-s single-channel EEG patches in order to minimize a regression objective in the spectral domain (i.e. on the amplitude and phase of the input patch). Second, a Transformer encoder is pretrained on a self-supervised task in which the model must predict the tokens that correspond to masked input patches. Three variants of the proposed model (with 5.8 up to 360M parameters) are pretrained on a diverse dataset of more than 2,500 h of EEG recordings and then finetuned on one of four supervised classification or regression downstream tasks. The proposed models outperform existing baselines on these tasks. Ablations on the amount of pretraining data, tokenization approach, masked prediction loss and masking support the proposed pretraining task and model configuration.

**Strengths:**

Originality: The proposed approach follows logically from existing work on pretraining Transformers on a corpus of combined EEG datasets, however the combination of a vector quantization tokenizer, a regression objective in the spectral domain and the pretraining on a very large set of EEG recordings appears novel.

Quality: The paper is of good quality, with strong core results showing the performance of the proposed approach, and multiple supporting analyses and ablation studies that support the methodological choices that were made. A few claims might not be completely supported by the results (see Weaknesses).

Clarity: The paper is overall clearly written, with mostly clear descriptions of the proposed approach and of the results. Some methodological points require clarification (see Weaknesses, first point).

Significance: Overall, this study is an important step towards bridging the gap between the approaches used in large language modeling and EEG processing. The results on “EEG scaling laws” (Section 3.6) are a first attempt at answering an important question in the field of deep learning and EEG.

**Weaknesses:**

- Some methodological points require clarification, e.g. the impact and choice of windowing/tokenization hyperparameters (Q1), the learning and reuse of spatial embeddings (Q3), the potential overlap between pretraining and downstream datasets (Q5) and the sampling of examples during pretraining (Q8).

- I don’t think the results are clear enough to deduce what is claimed in the analysis of Figure 5, i.e. that the performance saturates for the base model at 500 h and for the large model at 2000 h. For instance, the Large models seem to continue learning over 2000 h on TUEV. Similarly, the Base model might not be saturated yet; the performance curve seems pretty noisy. Maybe repeating this analysis with a log-scale would be clearer (1h, 10h, 100h, 1000h, 2500h).

- The use of the term “BCI” (e.g. in the title) is confusing as this typically refers to a subset of the tasks/datasets considered in this work. For instance, the term BCI is usually used to describe cases where there is an interface between brain activity and a computer that bypasses normal communication pathways. Under this definition, tasks such as pathology detection (TUAB) or event detection (TUEV) are not BCI tasks. I would recommend adapting the language of the manuscript to make this clearer.

**Questions:**

1. Some of the hyperparameter choices for the windowing and tokenization steps are not clear to me. First, what were the window strides ($s$ in first paragraph of Section 2.1) used for the different datasets? Tables 3 and 4 report a “Data stride” value but I’m not sure whether that’s the same thing. Second, what is the impact of the selected patch size and patch stride on performance, i.e. are these choices important? Related to the point about how symmetric masking might be providing regularization that is useful for larger models, would a smaller window and/or stride help create more pretraining examples?

2. It is not clear to me whether the weights of the temporal encoder from the tokenization step are reused in the pretraining step, or if only the architecture is the same. From the dimensions of the large and huge models I assume the weights could not be reused as the sizes are not the same.

3. My understanding is that new spatial embeddings must be learned from each montage that is seen (i.e. the $i$ in Equation 2). How many different spatial embeddings were learned during pretraining? Also, what spatial embeddings were used in the different ablations of Table 10 if the model didn’t have a chance to learn a spatial embedding for TUAB and TUEV (or were examples from this EEG montage already seen in the pretraining data)?

4. Section 2.3: What is the self-distillation loss term? It doesn’t seem to be in the final pretraining objective of Equation 12.

5. Is there an overlap between TUAB/TUEV and the different training sets taken from the Temple University EEG Corpus (TUAR, TUEP, etc.)? If so, could this explain why including TUAB and TUEV in the pretraining set didn’t change the results much (Section 3.5)? I believe this shouldn’t impact the comparison with BIOT as the reported results from the BIOT paper appear to be from the model also pretrained on TUAB and TUEV.

6. Related to the previous question, looking at Figure 4 it looks like adding TUEV to the pretraining dataset actually negatively impacts downstream performance. Is this effect significant and if so, what could be driving this decrease in performance?

7. Section 3.2 on preprocessing: the notch filtering at 50 Hz will not adequately remove power line interference in datasets collected in North America, such as the TUEG datasets. I expect results to improve if the authors correctly notch filter those datasets at 60 Hz instead.

8. How are the pretraining examples sampled? Since this is not described in the manuscript I would assume sequences were sampled uniformly across the entire training corpus, however I was wondering if the authors have considered more balanced sampling schemes, e.g. taking datasets, recordings and/or experimental paradigm-related information into account when sampling.

9. Where did the baseline results for Table 6 come from? They don’t seem to be in the BIOT paper.

10. A few typos:
- Figure 2: “Fuorier spectrum”
- Appendix J: “PRE-TRAINGING”
- Equation 14: Missing closing parenthesis

---

> ### Author Response · Authors · 2023-11-17
> **Response (1/3)**
>
> We are deeply grateful to you for the thorough review and constructive comments, which have greatly assisted us in improving the quality and presentation of our manuscript. Please see below for the point-by-point responses to your comments.
>
> > [**W1**] Some methodological points require clarification, e.g. the impact and choice of windowing/tokenization hyperparameters (Q1), the learning and reuse of spatial embeddings (Q3), the potential overlap between pretraining and downstream datasets (Q5) and the sampling of examples during pretraining (Q8).
>
> * We appreciate your concerns. In the following Questions section, we will address each aspect of your inquiry separately, providing a comprehensive response to ensure clarity and understanding.
>
> > [**W2**] I don’t think the results are clear enough to deduce what is claimed in the analysis of Figure 5, i.e. that the performance saturates for the base model at 500 h and for the large model at 2000 h. For instance, the Large models seem to continue learning over 2000 h on TUEV. Similarly, the Base model might not be saturated yet; the performance curve seems pretty noisy. Maybe repeating this analysis with a log-scale would be clearer (1h, 10h, 100h, 1000h, 2500h).
>
> * Thanks for your valuable suggestion. We have updated Figure 5 with additional experiments on 1h, 10h, and 100h. Please refer to Figure 5 in the revised paper. As can be seen from the trend in Figure 5, the performance of the Base model consistently increases with the amount of data at 500 hours or less. Therefore, 500 hours of EEG data is enough for pre-training the Base model. For the Large model, the performance tends to saturation and achieve its optimum as the data size reaches 2,000 hours.
>
> > [**W3**] The use of the term “BCI” (e.g. in the title) is confusing as this typically refers to a subset of the tasks/datasets considered in this work. For instance, the term BCI is usually used to describe cases where there is an interface between brain activity and a computer that bypasses normal communication pathways. Under this definition, tasks such as pathology detection (TUAB) or event detection (TUEV) are not BCI tasks. I would recommend adapting the language of the manuscript to make this clearer.
>
> * Thanks for your valuable opinion.  A Brain-Computer Interface (BCI) can be defined as "a system that measures central nervous system (CNS) activity and converts it into artificial output that replaces, restores, enhances, supplements, or improves natural CNS output, thereby altering the ongoing interactions between the CNS and its external or internal environment" [1]. Based on this definition, we propose that tasks like TUAB and TUEV contribute to the first part of the BCI system's measurement process. This involves measuring neural activity and utilizing it as input for subsequent processes, including control, intervention therapy, and neural circuit remodeling. Therefore, we contend that tasks such as TUAB and TUEV fall within the scope of BCI tasks.
>
> > [**Q1**] Some of the hyperparameter choices for the windowing and tokenization steps are not clear to me. First, what were the window strides (in first paragraph of Section 2.1) used for the different datasets? Tables 3 and 4 report a “Data stride” value but I’m not sure whether that’s the same thing. Second, what is the impact of the selected patch size and patch stride on performance, i.e. are these choices important? Related to the point about how symmetric masking might be providing regularization that is useful for larger models, would a smaller window and/or stride help create more pretraining examples?
>
> * We apologize for not clarifying some hyperparameters. For the window stride, it is exactly the same thing as the data stride in Table 3 and Table 4. As for the setting of patch size and patch stride, we set them to 1 second because 1 second is the smallest unit for many EEG tasks. The common time length for most EEG tasks is between 1 second and 10 seconds. For example, the time length of a sample for datasets we use in this paper is 10 seconds for TUAB, 5 seconds for TUEV, 1 second for SEED-V, and 2 seconds for MoBI. On the other hand, this setting is consistent with some baselines such as BIOT where the authors conduct experiments with different settings of patch size and overlapping length, and the results show that 1 second is a relatively good choice for EEG segments. For the setting of data stride, we set it to 4 seconds for pre-training based on the following considerations: (1) covering all training data. As the maximum sequence length is 256, the sample for 64-channel EEG signals is 4 seconds. All data can be visited if we set the data stride to 4 seconds (no overlapping). (2) boosting the training speed (reducing the number of samples). If we set the data stride to 1 second or less, the number of samples could be so large that it will take several times as long to complete the training.

---

> > ### Author Response · Authors · 2023-11-17
> > **Response (2/3)**
> >
> > > [**Q2**] It is not clear to me whether the weights of the temporal encoder from the tokenization step are reused in the pretraining step, or if only the architecture is the same. From the dimensions of the large and huge models I assume the weights could not be reused as the sizes are not the same.
> >
> > * The weights of the temporal encoder from the tokenization step are not reused in the pretraining step. It is just as what you say that only the architecture is the same.
> >
> > > [**Q3**] My understanding is that new spatial embeddings must be learned from each montage that is seen (i.e. the in Equation 2). How many different spatial embeddings were learned during pretraining? Also, what spatial embeddings were used in the different ablations of Table 10 if the model didn’t have a chance to learn a spatial embedding for TUAB and TUEV (or were examples from this EEG montage already seen in the pretraining data)?
> >
> > * First of all, our spatial embeddings are initialized according to the international 10-20 system (https://en.wikipedia.org/wiki/10%E2%80%9320_system_(EEG)) which was commonly followed by most existing EEG datasets. Though different datasets may have different configurations, they all utilize a subset of electrodes of the international 10-20 system. Another reason why we collect a variety of datasets with different configurations is that we hope LaBraM can see as many different electrodes as possible in the pre-training process. Consequently, examples from the EEG montage of TUAB and TUEV are already seen in the pretraining data. However, if there are unseen electrodes in the downstream dataset such as MoBI, we just randomly initialize new spatial embeddings for the unseen electrodes. The new spatial embeddings are trained along with the existing spatial embeddings throughout the fine-tuning process.
> >
> > > [**Q4**] Section 2.3: What is the self-distillation loss term? It doesn’t seem to be in the final pretraining objective of Equation 12.
> >
> > * We apologize for this mistake. This is a clerical error and we have removed it in the revised paper.
> >
> > > [**Q5**] Is there an overlap between TUAB/TUEV and the different training sets taken from the Temple University EEG Corpus (TUAR, TUEP, etc.)? If so, could this explain why including TUAB and TUEV in the pretraining set didn’t change the results much (Section 3.5)? I believe this shouldn’t impact the comparison with BIOT as the reported results from the BIOT paper appear to be from the model also pretrained on TUAB and TUEV.
> >
> > * There is no overlap between TUAB/TUEV and the different training sets taken from the Temple University EEG Corpus (https://isip.piconepress.com/projects/tuh_eeg/html/downloads.shtml) and they are disjoint from TUAR, TUEP, TUSZ, and TUSL. Since we pre-train LaBraM with tremendous EEG data, the minor differences when including TUAB/TUEV demonstrate that our model has the capability to learn universal EEG representations.
> >
> > > [**Q6**] Related to the previous question, looking at Figure 4 it looks like adding TUEV to the pretraining dataset actually negatively impacts downstream performance. Is this effect significant and if so, what could be driving this decrease in performance?
> >
> > * The large-scale pre-training has brought about the strong capability for learning generic representations, which might be saturated even if we further utilize TUEV. Thus, the slight decrease on TUEV might also be due to the randomness of the pre-training process. Considering ablation on both TUAB and TUEV, we observe no obvious differences between pre-training with and without downstream tasks.
> >
> > > [**Q7**] Section 3.2 on preprocessing: the notch filtering at 50 Hz will not adequately remove power line interference in datasets collected in North America, such as the TUEG datasets. I expect results to improve if the authors correctly notch filter those datasets at 60 Hz instead.
> >
> > * Thanks for your valuable suggestion. We acknowledge this is our oversight. EEG signals usually fall into the low-frequency range, which are divided into bandwidths known as delta (0.5-4 Hz), theta (4-8 Hz), alpha (8-14 Hz), beta (14-30 Hz), and gamma (30-50 Hz) [2]. Different frequency bands are associated with different states of brain activities, so most neuron activities are within the low-frequency range (<50 Hz). Therefore, frequencies higher than 50 Hz contribute less to EEG decoding, from which we think the notch filtering might not be so significant for the results. Nevertheless, we deeply appreciate your suggestion which could indeed improve the quality of our paper.

---

> > > ### Author Response · Authors · 2023-11-17
> > > **Response (3/3)**
> > >
> > > > [**Q8**] How are the pretraining examples sampled? Since this is not described in the manuscript I would assume sequences were sampled uniformly across the entire training corpus, however I was wondering if the authors have considered more balanced sampling schemes, e.g. taking datasets, recordings and/or experimental paradigm-related information into account when sampling.
> > >
> > > * For simplicity, the pre-training samples are uniformly sampled within a set of datasets that are of the same EEG montage. The order of the set of datasets is also random. This might have the potential to improve if we can design a more advanced sampling strategy and is indeed a promising direction for future work.
> > >
> > > > [**Q9**] Where did the baseline results for Table 6 come from? They don’t seem to be in the BIOT paper.
> > >
> > > * These results are replicated by ourselves using the codes provided by the authors of BIOT.
> > >
> > > > [**Q10**] A few typos
> > >
> > > * Thanks a lot for your careful check and we have corrected the typos in the revised paper.
> > >
> > > **Reference**
> > >
> > > [1] Wolpaw J R, Wolpaw E W. Brain-computer interfaces: something new under the sun[J]. Brain-computer interfaces: principles and practice, 2012, 14.
> > >
> > > [2] Walter D O. Spectral analysis for electroencephalograms: mathematical determination of neurophysiological relationships from records of limited duration[J]. Experimental Neurology, 1963, 8(2): 155-181.

---

> > > > ### Comment · Reviewer_WxGC · 2023-11-21
> > > >
> > > > Thanks to the authors for their answers and clarifications.
> > > >
> > > > [W2] Thank you for updating the figure. I believe my point still holds however: it’s not clear the performance of any of the models has saturated yet (with the exception of the base model evaluated on TUAB) as the curves are too noisy to conclude that performance wouldn’t improve with more data. I think it would be clearer and more precise to say that e.g. XX% of the 2,500-hour performance can be achieved with 500 h of data for the base model (and similarly for the large model, etc.).
> > > >
> > > > [W3] I am not entirely convinced that e.g. the pathology detection task on TUAB fits with the definition you provide. The last part “altering the ongoing interactions between the CNS and its external or internal environment” doesn’t seem to apply to this task (unless you assume the clinical/hospital setting to be the environment the patient is interacting with, which might be stretching it). In any case, I don’t think this choice of vocabulary has an important impact on the submission - it might just reduce the reach of the paper.

---

> > > > > ### Author Response · Authors · 2023-11-22
> > > > >
> > > > > [W2] Thank you for your further feedback, and we are convinced by your meticulous observations. In light of your suggestions, we have revised the analysis in our paper as follows: "As illustrated in Figure 5, the performance of the Base model with 500 hours of training exceeds that of the 2500-hour model on TUAB, while approaching over 90% of the 2500-hour performance on TUEV. For the Large model, performance generally improves with increased data volume, though the growth rate slows after 1000 hours. In contrast, the Huge model exhibits a noticeable upward trend in performance as data size continues to expand." We sincerely appreciate your constructive suggestion, which has greatly enhanced the credibility of our paper.
> > > > >
> > > > > [W3]  Thanks for your valuable opinion.  Given the constraints on the title and abstract, we will indeed exercise caution in our choice of vocabulary in future studies. We sincerely appreciate your valuable suggestions, which will enable us to be more rigorous in our future research endeavors.

---

### Official Review · Reviewer_tjUb · 2023-11-05

**Soundness:** 3 good
**Presentation:** 3 good
**Contribution:** 3 good
**Rating:** 6
**Confidence:** 3

**Summary:**

The paper proposes a method and model for self-supervised training on large-scale EEG data coming from various datasets with different electrode configurations. First they train a neural tokenizer which learns to compress the EEG signal into vector-quantized encodings and reconstruct the amplitude and phase of the EEG signal from those. Then, given the trained neural tokenizer, a model is trained to reconstruct the vector quantized encodings of an EEG signal decoded from a masked version of the same encodings.  After these pretraining phases, the model is finetuned and evaluated on a downstream task. The authors report improved decoding accuracy on pathology diagnosis and clinical EEG event type classification compared to published work.

**Strengths:**

* Interesting method for self-supervised learning from EEG data
* Large collection of publicly available datasets
* Improved results over other self-supervised methods
* Analysis of scaling behavior

**Weaknesses:**

1)
Some papers have not been mentioned that work on heterogeneous datasets:
* [Learning Topology-Agnostic EEG Representations with Geometry-Aware Modeling](https://openreview.net/forum?id=hiOUySN0ub)
* [EEG Decoding for Datasets with Heterogenous Electrode Configurations using Transfer Learning Graph Neural Networks](https://arxiv.org/abs/2306.13109v1)
* [Generalizable Movement Intention Recognition with Multiple Heterogeneous EEG Datasets](https://ieeexplore.ieee.org/document/10160462)

2)
The choice to use mean squared error on phase values is strange to me. Due to their cyclical nature, very nearby phases, e.g., -pi+eps,pi-eps would get a large squared error that would also depend on whether one uses phases from 0 to 2pi or -pi to pi etc. So would make more sense tome to either always only use the minimum distance, so if one would put the phases on a unit circle the minimum distance on the circle, or maybe even regress fourier coefficients instead of amplitude/phase. I even wonder what would happen if one just does not put any loss on the phase prediction at all, only on the amplitudes that would be interesting to check as well.


3)
Why bold lowest std in table 1 and 2, better remove that, is rather confusing

4)
Font could be a bit bigger in Figure 1 and also parts of Girue 2 (e.g., channel names on bottom)
Fig 2 the amplitude phase plot on top right is confusing to me.
Symmetric in Figure 2 not symmetric

**Questions:**

1)
I assume Table 1/2 only compares to other self-supervised models? That should be written a bit more explicitly otherwise there are other papers worth citing:

https://www.springerprofessional.de/en/chrononet-a-deep-recurrent-neural-network-for-abnormal-eeg-ident/16824220
https://www.sciencedirect.com/science/article/pii/S1053811920305073
Might be in any case good to also show a purely supervised baseline.

2)
Are TUAB and TUEV disjoint recordingwise from TUAR TUEP TUSZ and TUSL?

---

> ### Author Response · Authors · 2023-11-17
>
> We are deeply grateful to you for the thorough review and constructive comments, which have greatly assisted us in improving the quality and presentation of our manuscript. Please see below for the point-by-point responses to your comments.
>
> > [**W1**] Some papers have not been mentioned that work on heterogeneous datasets
>
> * Thank you for your insightful suggestions. We appreciate your encouragement to broaden our study's scope. In response, we have integrated these recommendations into our revised paper, specifically in the related work section.
>
> > [**W2**] The choice to use mean squared error on phase values is strange to me. Due to their cyclical nature, very nearby phases, e.g., -pi+eps,pi-eps would get a large squared error that would also depend on whether one uses phases from 0 to 2pi or -pi to pi etc. So would make more sense tome to either always only use the minimum distance, so if one would put the phases on a unit circle the minimum distance on the circle, or maybe even regress fourier coefficients instead of amplitude/phase. I even wonder what would happen if one just does not put any loss on the phase prediction at all, only on the amplitudes that would be interesting to check as well.
>
> * Thank you for your insightful comments on the selection of mean squared error for phase values. We acknowledge your concerns about the cyclical nature of phases and the potential biases associated with different ranges. We are currently making efforts to conduct additional experiments based on this suggestion. However, due to limited computational resources and time constraints, we may not be able to complete these experiments in time. Rerunning the neural tokenizer training and LaBraM pre-training (especially with a total of 2,500 hours of data) can take several weeks. If the experiments are completed, we will update the results as promptly as possible. We believe that our comprehensive analysis will help researchers better understand the merits of each method and their applicability in different scenarios. We sincerely appreciate your constructive feedback, which has led to a more in-depth exploration of our research. Your insights have enriched our paper and improved its overall quality.
>
> > [**W3**] Why bold lowest std in table 1 and 2, better remove that, is rather confusing
>
> * Thank you for your suggestion, and we have made the necessary changes accordingly.
>
> > [**W4**] Font could be a bit bigger in Figure 1 and also parts of Girue 2 (e.g., channel names on bottom) Fig 2 the amplitude phase plot on top right is confusing to me. Symmetric in Figure 2 not symmetric
>
> * Thank you for your valuable feedback. We have taken your suggestions into account and made the necessary adjustments to improve the clarity and readability of the figures.
>
> > [**Q1**] I assume Table 1/2 only compares to other self-supervised models? That should be written a bit more explicitly otherwise there are other papers worth citing
>
> * Thanks for your valuable suggestion. We appreciate your attention to the details of the baselines presented in Tables 1 and 2. To provide a clearer understanding of these baselines, we present the following detailed descriptions:
>
> 1) SPaRCNet: This is a 1D-CNN-based model equipped with dense residual connections.
>
> 2) ContraWR: It converts biosignals into multi-channel spectrograms and then employs a 2D-CNN-based ResNet framework accompanied by contrastive learning.
>
> 3) CNN-Transformer: This hybrid model combines the strengths of CNNs and Transformers for enhanced performance.
>
> 4) FFCL: The Fine-tuning Framework for Classification (FFCL) integrates embeddings from both CNN and LSTM encoders for effective feature fusion.
>
> 5) ST-Transformer: This multi-level EEG transformer concurrently learns spatial and temporal features for improved performance.
>
> 6) BIOT: A framework based on the Transformer architecture, it takes spectral energy vectors extracted by FFT on the original signals as inputs. It can be pre-trained using either supervised or self-supervised learning techniques.
>
> It is worth noting that these baselines represent state-of-the-art supervised (SPaRCNet, CNN-Transformer, FFCL, and ST-Transformer) and self-supervised (ContraWR and BIOT) methods in the recent literature. We have sincerely cited the mentioned paper in our revised manuscript, specifically in the introduction section. We hope this provides clarification and addresses your concerns.
>
> > [**Q2**] Are TUAB and TUEV disjoint recordingwise from TUAR TUEP TUSZ and TUSL?
>
> * Yes, the TUAB and TUEV are indeed disjoint from TUAR, TUEP, TUSZ, and TUSL. Detailed descriptions of these datasets can be found in the provided URL link (https://isip.piconepress.com/projects/tuh_eeg/html/downloads.shtml).

---

> ### Comment · Reviewer_tjUb · 2023-11-20
>
> Thanks for your comments.
> It seems you report per-10-second-sample balanced accuracy on TUAB and that is where the discrepancy comes from to other results, e.g., in https://www.sciencedirect.com/science/article/pii/S1053811920305073 (best recordingwise accuracy is around 86%, if one computes balanced accuracy for TCN in Figure 7, it is 85.3%). Please additionally report recordingwise accuracy by averaging the predictions of all 10 second samples that belong to the same recording.

---

> ### Author Response · Authors · 2023-11-21
>
> Thank you for your prompt response. We appreciate your concern and are pleased to offer the recording-wise accuracy of TUAB by averaging the predictions of all 10-second samples belonging to the same recording: **84.08%** (for the Base model), which is 2.68% higher than the balanced accuracy reported in our paper.  Regarding the reason for not conducting a recording-wise comparison at the outset, it is crucial to recognize that the outcomes presented in our paper are derived under identical experimental configurations, facilitating horizontal comparisons. As you are aware, various dataset divisions and experimental arrangements can result in non-comparable findings. For instance, in the paper you mentioned [1], their basic settings differ a lot from ours:
>
> 1) The data division is very different. [1] performed 5-fold cross-validation without test set while we splitted the dataset with training, validation, and test set. It is noted that the test set is provided by the dataset itself, so we just divided the training patients into training and validation groups by 80% and 20%.
>
> 2) [1] discarded the first 60 seconds of every recording while we did not. This would bring a higher accuracy because they observed a large number of recording artifacts in this period.
>
> 3) [1] used a maximum of 20 min of every recording to avoid considerable feature generation and resampling times for exceptionally long recordings while we did not.
>
> 4) [1] resampled the EEG signals to 100 Hz while we resampled them to 200 Hz.
>
> 5) The length of a sample in [1] was 6 seconds while we utilized 10-second samples.
>
> Although discrepancies may exist between recording-wise accuracy and balanced accuracy, we believe that both metrics offer a direct reflection of the model's performance and share a consistent change trend. Nevertheless, we appreciate your insight as it can enrich and enhance the credibility of our paper's conclusions.
>
> **Reference**
>
> [1] Gemein L A W, Schirrmeister R T, Chrabąszcz P, et al. Machine-learning-based diagnostics of EEG pathology[J]. NeuroImage, 2020, 220: 117021.

---

### Meta-Review · Area_Chair_xkTN · 2023-12-14

**Metareview:**

This paper presents the Large Brain Model (LaBraM), a single large neural network aimed at being a universal EEG model. The authors accomplish this by leveraging large language models (LLMs) trained on tokenized versions of the EEG data. Specific tasks then use this general "Large EEG Model" by fine-tuning for specific tasks. The authors train their model on large amounts of diverse data, and demonstrate improvements in multiple tasks. The reviewers universally appreciated the capabilities of the model and the design of mixing the vector-quanitization and LLM for this application. While some concerns were raised, such as the effect of training on the MSE to a cyclical variable (phase) and clarity/clarification issues, these remain minor concerns. Therefore I recommend this paper be accepted.

**Justification For Why Not Higher Score:**

While the paper was good, there were a few more minor points that prevented a higher talk, including the training on the MSE to a cyclical variable (phase) and clarity/clarification issues.

**Justification For Why Not Lower Score:**

The potential impact of large-scale "universal" models for time-series data outside of language stands to be impactful. Specifically in neuroscience it's not clear that the diversity of recording equipment and lab-to-lab protocols would permit such a model, as is demonstrated here.

---

### Decision · Program_Chairs · 2024-01-16

Accept (spotlight)